# Lattice Bose polarons at strong coupling and quantum criticality

Ragheed Alhyder[1,2⋆], Victor E. Colussi[3,4], Matija Čufar[5,6],
Joachim Brand[5,6], Alessio Recati[3†] and Georg M. Bruun[2]

⋆ ragheed.alhyder@ist.ac.at , † alessio.recati@ino.cnr.it

## Abstract

The problem of mobile impurities in quantum baths is of fundamental importance in many-body physics. There has recently been significant progress regarding our understanding of this due to cold atom experiments, but so far it has mainly been concerned with cases where the bath has no or only weak interactions, or the impurity interacts weakly with the bath. Here, we address this gap by developing a new theoretical framework for exploring a mobile impurity interacting strongly with a highly correlated bath of bosons in the quantum critical regime of a Mott insulator (MI) to superfluid (SF) quantum phase transition. Our framework is based on a powerful quantum Gutzwiller (QGW) description of the bosonic bath combined with diagrammatic field theory for the impurity-bath interactions. By resumming a selected class of diagrams to infinite order, a rich picture emerges where the impurity is dressed by the fundamental modes of the bath, which change character from gapped particle-hole excitations in the MI to Higgs and gapless Goldstone modes in the SF. This gives rise to the existence of several quasiparticle (polaron) branches with properties reflecting the strongly correlated environment. In particular, one polaron branch exhibits a sharp cusp in its energy, while a new ground-state polaron emerges at the $O(2)$ quantum phase transition point for integer filling, which reflects the nonanalytic behavior at the transition and the appearance of the Goldstone mode in the SF phase. Smooth versions of these features are inherited in the polaron spectrum away from integer filling due to the influence of Mott physics on the bosonic bath. We furthermore compare our diagrammatic results with quantum Monte Carlo calculations, obtaining excellent agreement. This accuracy is quite remarkable for such a highly non-trivial case of strong interactions between the impurity and bosons in a maximally correlated quantum critical regime, and it establishes the utility of our framework. Finally, our results show how impurities can be used as quantum sensors and highlight fundamental differences between experiments performed at a fixed particle number or a fixed chemical potential.

1 Institute of Science and Technology Austria (ISTA),
Am Campus 1, 3400 Klosterneuburg, Austria
2 Center for Complex Quantum Systems, Department of Physics and Astronomy,
Aarhus University, Ny Munkegade 120, DK-8000 Aarhus C, Denmark
3 Pitaevskii BEC Center, CNR-INO and Dipartimento di Fisica,
Università di Trento, I-38123 Trento, Italy
4 Infleqtion Inc., 3030 Sterling Circle, Boulder, Colorado 80301 USA

**5** Te Whai Ao — Dodd-Walls Centre for Photonic and Quantum Technologies,
Auckland 0632, New Zealand
**6** Centre for Theoretical Chemistry and Physics, New Zealand Institute for Advanced Study,
Massey University, Private Bag 102904, North Shore, Auckland 0745, New Zealand

## Contents

# 1 Introduction

The properties of dilute mobile impurities that interact with a quantum environment play a key role in our understanding of nature. The problem is closely connected with the concept of quasiparticles, which provides a relatively simple but very powerful description of many-body systems [1]. Indeed, an impurity particle smoothly evolves into a quasiparticle when interactions with its environment are turned on, assuming that no phase transitions occur. This quasiparticle consists of the bare impurity dressed by excitations in its environment– a picture that first emerged when studying an electron that excited lattice vibrations (phonons) in a dielectric, thereby forming a so-called polaron [2,3]. Since then, quasiparticles have been used successfully to describe a wide range of quantum many-body systems, from liquid helium mixtures and electrons in crystals, to nuclear matter and quark-gluon plasmas [4–6].

Our understanding of mobile impurities in quantum environments and the formation of quasiparticles has improved dramatically in recent years, mostly driven by impressive experiments with cold atomic gases. These experiments have mainly focused on impurities in ideal or weakly interacting Fermi gases forming so-called Fermi polarons [7,8], and more recently also on impurities in weakly interacting Bose-Einstein condensates (BECs) forming Bose polarons [9]. The Bose polaron consists of the impurity dressed by the Goldstone (Bogoliubov sound) modes of the BEC, which, as we shall see, is a limiting case of the problem considered in the present work.

Less attention has been devoted to mobile impurities and polaron formation in the case when the bath is strongly correlated. Theoretical investigations include mobile impurities in fermionic superfluids [10–14], in a lattice gas of hard core bosons [15], in supersolids [16], in Bose systems in one dimension [17], and in topological systems [18–23]. The case of an impurity interacting with a bath in the vicinity of a quantum critical point is particularly interesting because there are strong quantum fluctuations across many length scales, which, however, also makes it very challenging. Recently, an infinitely heavy impurity in a fermionic system undergoing a Mott insulator to metal transition was investigated [24].

The Bose-Hubbard (BH) model describing repulsively interacting bosons in a lattice realizes at integer filling a prototypical case of a quantum phase transition: the bosons are in a Mott insulator (MI) phase for small hopping and in a superfluid (SF) phase for large hopping with quantum phase transitions in the $O(2)$ universality class in between [25].[1] In an early study, a mobile impurity immersed in such a BH model was explored using effective field theory [26]. Recently, perturbation theory was used to explore a mobile impurity across the MI-SF transition [27]. Here, the quantum Gutzwiller (QGW) approach, which was shown to accurately capture features related to the quantum fluctuations and the static and dynamical properties of the BH model in Refs. [28,29], was used to elucidate the fundamental properties of the resulting polaron in the quantum critical region. Previous work with the QGW method also considered a fixed impurity weakly coupled to a BH bath as a sensor for the MI-SF transition able to identify the different universality classes of the model [30].

In this paper, we explore the properties of a mobile impurity interacting with repulsively interacting bosons at integer filling in a square lattice. Using a QGW description of the bosons in the quantum critical regime of the MI to SF transition, we develop a field theory describing how the impurity excites elementary excitations of the bosons as it moves through the lattice. We then apply a generalized ladder approximation, which includes strong two-body correlations and the formation of impurity-boson bound states. Using this, we show that the interplay between strong impurity-boson interactions and boson-boson correlations in the quantum-critical regime gives rise to rich physics with several polaron branches. At the MI-SF phase-

---

[1]At non-integer filling the phase transition is due to a filling change. It is known as commensurate-incommensurate phase transition and is essentially a vacuum to matter Bose-Einstein condensation [25].

transition point, one polaron branch exhibits a cusp in its energy, and a new ground-state polaron appears. These distinctive features arise because both the Goldstone and the Higgs modes become gapless at the phase transition, and they also appear in smoothed-out versions for filling fractions slightly different from unity. Our results are shown to agree remarkably well with projector quantum Monte-Carlo (QMC) calculations for the many-body ground state, which demonstrates the usefulness of our theoretical framework for describing this strongly correlated many-body problem. Finally, we discuss important differences between performing experiments at constant particle number or at constant chemical potential.

## 2 System

We consider a system consisting of a single mobile impurity interacting with a bath of bosons in a square lattice composed of $M$ sites as described by the BH model. The Hamiltonian reads

$$\hat{H} = -t \sum_{\langle r,s \rangle} (\hat{a}_r^\dagger \hat{a}_s + \hat{c}_r^\dagger \hat{c}_s + \text{h.c.}) - \mu \sum_r \hat{n}_r + \frac{U}{2} \sum_r \hat{n}_r(\hat{n}_r - 1) + U_{\text{IB}} \sum_r \hat{n}_{I,r} \hat{n}_r, \qquad (1)$$

where $\hat{a}_r^\dagger$ ($\hat{c}_r^\dagger$) are creation operators for bosons (the impurity) on lattice site $r$. The first line describes the hopping of the bosons and the impurity with equal tunneling parameter $t$ for simplicity, and $\mu$ is the boson chemical potential. The energy dispersion of both particles in the absence of interactions is $\varepsilon_{\boldsymbol{k}} = 4t[\sin^2(k_x/2) + \sin^2(k_y/2)]$ where we have used the usual lattice Fourier transform and taken the minimum to define zero energy. We use units where the lattice constant is unity. The second line in Eq. (1) describes the onsite interaction between the bosons with strength $U > 0$ and between the impurity and the bosons with strength $U_{\text{IB}}$, where $\hat{n}_{\mathbf{r}} = \hat{a}_r^\dagger \hat{a}_r$ and $\hat{n}_{I,\mathbf{r}} = \hat{c}_r^\dagger \hat{c}_r$. Depending on the ratio $t/U$ and the chemical potential of the bosons, they form either a MI or SF giving rise to the phase diagram shown in Fig. 1(a). We will in the following explore the properties of the impurity in the region around the $O(2)$ phase transition between the MI and the SF at integer filling.

## 3 Methods

Previously, the lattice polaron in the transition region was studied for weak impurity-bath interaction ($|U_{\text{IB}}/U| \ll 1$) in two dimensions using perturbative methods within the QGW approach [27]. In Sec. 3.1, basics of the QGW method are recapped followed by the development of our novel diagrammatic theory for the impurity-bath system, which is capable of describing the regime of strong interactions not only between the bosons but also between the impurity and the bosons. In Sec. 3.2, we describe our Quantum Monte-Carlo calculations.

### 3.1 Quantum Gutzwiller (QGW) method

In this section, we develop a novel strong interaction approach to describe mobile impurities immersed in a BH bath in the quantum critical regime. The method is based on combining the semi-analytic QGW method with diagrammatic field theory, which we will implement in practice using a generalized self-consistent ladder approximation. We begin with a recount of the QGW method before outlining the diagrammatic technique. Further details on the QGW method and numerical implementation can be found in App. A.

    The Gutzwiller ansatz for the ground state of the BH Hamiltonian $|\Psi_G^0\rangle = \bigotimes_r \sum_n c_n^0 |n, r\rangle$ is a tensor product of lattice Fock states $|n, r\rangle$ with amplitude $c_n^0$ describing the occupation of site $r$ by $n$ bosons. Within this mean-field treatment, the bath density is given by $n_0 = \sum_n n |c_n^0|^2$, and

the condensate order parameter by $\psi_0 = \sum_n \sqrt{n} c_{n-1}^0 c_n^0$. Minimizing the energy with respect to $|\Psi_G^0\rangle$ produces the phase diagram shown in Fig. 1(a). We refer the reader to Ref. [31] (and references therein) for further details of the ground state calculation.

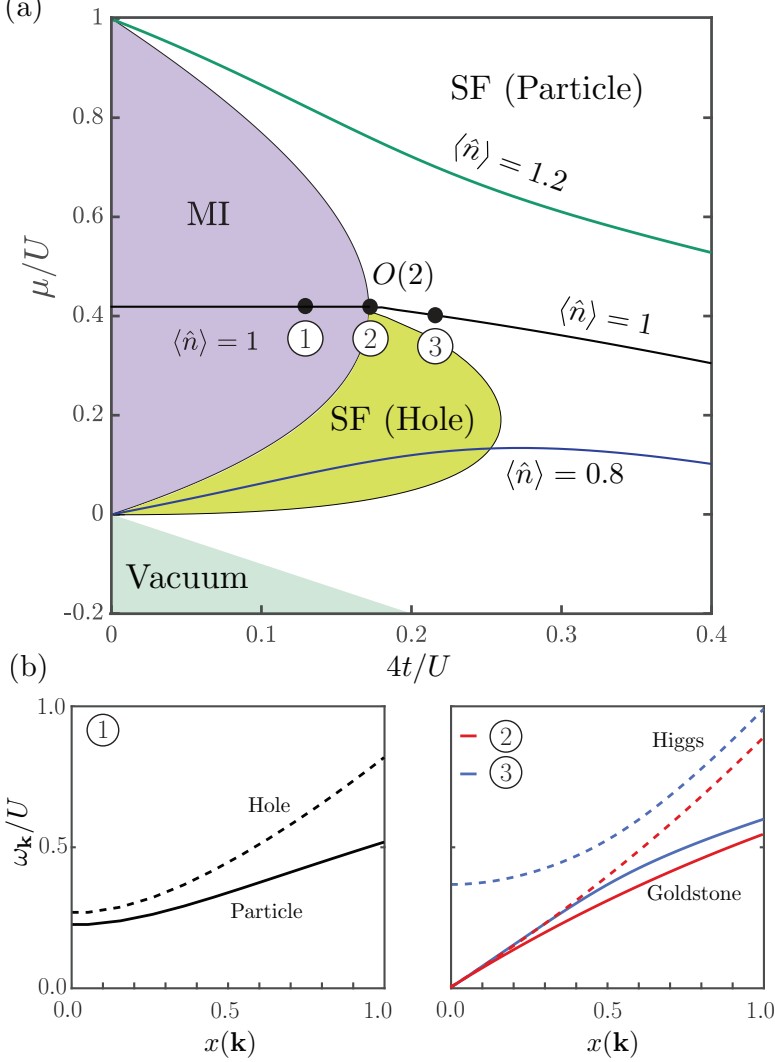

Figure 1: (a) Phase diagram of repulsively interacting bosons as a function of $t/U$ and the chemical potential as calculated using the mean-field Gutzwiller approach for the 2D case considered in this work; see [31] and references thererin. The shaded purple area indicates the Mott insulator lobe of unit filling while the shaded green area indicates the region of the hole superfluid. Lines of constant filling are shown and ①, ② and ③ indicate points where the polaron properties are explored in detail in Sec. 4. (b) In the bottom two panels are shown the lowest (solid) and second lowest (dashed) elementary excitations of the bosons at the points in the phase diagram considered in this work versus $x(\mathbf{k}) = \sqrt{\epsilon_{\mathbf{k}}/8t}$ which varies from 0 to 1, and $x \approx |\mathbf{k}|/2\sqrt{2}$ for small $|\mathbf{k}|$. These modes are gapped particle-hole excitations in the MI, gapless Goldstone and Higgs modes at the $O(2)$ phase transition, and gapless Goldstone and gapped Higgs modes in the SF.

The QGW method describes density fluctuations on top of the ground state $|\Psi_G^0\rangle$ by canonical quantization as $\delta \hat{c}_n(\mathbf{r}) = M^{-1/2} \sum_{\lambda \mathbf{k}} e^{i\mathbf{k}\cdot\mathbf{r}} (u_{\lambda,\mathbf{k},n} \hat{b}_{\lambda,\mathbf{k}} + v_{\lambda,\mathbf{k},n} \hat{b}_{\lambda,-\mathbf{k}}^\dagger)$ [28, 32, 33]. Here, $\hat{b}_{\lambda,\mathbf{k}}^\dagger$ ($\hat{b}_{\lambda,\mathbf{k}}$) is a bosonic operator that creates (annihilates) an elementary excitation of the bosons in the $\lambda^{\text{th}}$ branch with momentum $\mathbf{k}$, energy $\omega_{\lambda,\mathbf{k}}$, and coefficient $v_{\lambda,\mathbf{k},n}$ ($u_{\lambda,\mathbf{k},n}$), which can be chosen to be real [34]. Deep in the MI phase this corresponds to an individual particle or hole, while the lowest mode corresponds to the Bogolioubov sound deep in the SF phase. Expanding the bath Hamiltonian to quadratic order in the fluctuations yields

$$\hat{H}_{\text{B}} = \sum_{\lambda,\mathbf{k}} \omega_{\lambda,\mathbf{k}} \hat{b}_{\lambda,\mathbf{k}}^\dagger \hat{b}_{\lambda,\mathbf{k}}, \tag{2}$$

which describes a set of non-interacting bosonic modes. The QGW approach leading to Eq. (2) is detailed in [28] where it was shown to provide a remarkably accurate description of the bosonic bath even in critical regimes where quantum fluctuations are strong. Further details of the QGW approach are provided in App. A.1.

In this work, we consider the BH bath at (or very close to) integer filling in the region of the phase diagram where the MI-SF transition belongs to the $O(2)$ universality class, as indicated in Fig. 1(a) [25]. Here, the bath can be described by an effective relativistic (Lorentz-invariant) field theory due to particle-hole symmetry. This symmetry yields a decoupling of amplitude and phase degrees of freedom, resulting in a gapless Goldstone and a massive so-called Higgs mode on the superfluid side ($\psi_0 \neq 0$) and gapped particle and hole excitations on the Mott side of the transition. The mass of the Higgs mode vanishes at the transition point as shown in Fig. 1(b) [34–38]. The superfluid at the $O(2)$ point is of *hybrid* particle-hole character as the derivative $\mu'(t) \equiv \partial \mu / \partial t$ evaluated at constant filling vanishes, separating a bounded region of hole superfluidity $\mu'(t) < 0$ from the (unbounded) region of particle superfluidity, as shown in Fig. 1(a) [31]. As discussed in the seminal work [39], the hole or particle characterization of the superfluid close to the Mott lobe is made in analogy with electron or hole conductance of a material in solid-state physics. Ramifications for the character of vortices, solitons, and elementary excitations are considered in Refs. [34, 40, 41]. Notably, the region of particle superfluidity extends into the deep superfluid region where the lattice polaron in a weakly interacting BEC was previously considered [42].

### 3.1.1 Diagrammatics

We now use the QGW formalism as a foundation to construct our diagrammatic field theory for analyzing the properties of a mobile impurity interacting with the strongly correlated bosonic bath.

In this approach, the basic object to calculate is the impurity Green's function $G(\mathbf{q}, \omega)$, and we start by expanding the impurity-bath interaction $\hat{H}_{\text{IB}}$ into different processes involving the impurity and excitations of the bath. This expansion was given in Ref. [27], and here we reformulate it in a way suitable for establishing the Feynman rules of a diagrammatic theory capable of describing strong interactions. Expanding the impurity-bath interaction to quadratic order in bath fluctuations yields

$$\hat{H}_{\text{IB}} \approx U_{\text{IB}} \sum_{\mathbf{r}} \hat{n}_{I,\mathbf{r}} [n_0 + \delta_1 \hat{n}(\mathbf{r}) + \delta_2 \hat{n}(\mathbf{r}) + \dots], \tag{3}$$

where the first term is the bath density in the mean-field ground state, while the second and third terms are linear and quadratic in the bath fluctuation operators, respectively. This ex-



pansion reads in momentum space (see Sec. A.1 for details)

$$\hat{H}_{IB} = U_{IB} \sum_{\boldsymbol{k},\boldsymbol{p},\boldsymbol{q}} \sum_{\lambda,\lambda'} \Bigg[ U_{\lambda\boldsymbol{k},\lambda'\boldsymbol{p}} \hat{c}_{\boldsymbol{q}}^{\dagger} \hat{c}_{\boldsymbol{q}-\boldsymbol{p}+\boldsymbol{k}} \hat{b}_{\boldsymbol{k},\lambda}^{\dagger} \hat{b}_{\boldsymbol{p},\lambda'} + V_{\lambda\boldsymbol{k},\lambda'\boldsymbol{p}} \hat{c}_{\boldsymbol{q}}^{\dagger} \hat{c}_{\boldsymbol{q}+\boldsymbol{p}-\boldsymbol{k}} \hat{b}_{\boldsymbol{k},\lambda} \hat{b}_{\boldsymbol{p},\lambda'}^{\dagger}$$
$$+ \frac{1}{2} W_{\lambda\boldsymbol{k},\lambda'\boldsymbol{p}} \left( \hat{c}_{\boldsymbol{q}}^{\dagger} \hat{c}_{\boldsymbol{q}+\boldsymbol{p}+\boldsymbol{k}} \hat{b}_{\boldsymbol{k},\lambda}^{\dagger} \hat{b}_{\boldsymbol{p},\lambda'}^{\dagger} + \hat{c}_{\boldsymbol{q}}^{\dagger} \hat{c}_{\boldsymbol{q}-\boldsymbol{p}-\boldsymbol{k}} \hat{b}_{\boldsymbol{k},\lambda} \hat{b}_{\boldsymbol{p},\lambda'} \right) \Bigg],$$

(4)

with two-particle vertices

$$U_{\lambda\boldsymbol{k},\lambda'\boldsymbol{q}} = \sum_{n} \Big[ n - n_0(1 - \delta_{\lambda\lambda'}\delta_{\lambda,0}) \Big] u_{\boldsymbol{k},n,\lambda} u_{\boldsymbol{q},n,\lambda'}^{*},$$
$$V_{\lambda\boldsymbol{k},\lambda'\boldsymbol{q}} = \sum_{n} (n - n_0) v_{\boldsymbol{k},n,\lambda} v_{\boldsymbol{q},n,\lambda'}^{*},$$

(5)

$$W_{\lambda\boldsymbol{k},\lambda'\boldsymbol{q}} = \sum_{n} (n - n_0)(u_{\boldsymbol{k},n,\lambda} v_{\boldsymbol{q},n,\lambda'}^{*} + u_{\boldsymbol{q},n,\lambda'}^{*} v_{\boldsymbol{k},n,\lambda}),$$

corresponding to the structure factors of the density channel [30]. Here, the Gutzwiller mean-field ground state is, for convenience, denoted by the $\lambda = 0$ "mode" with energy $\omega_{\lambda=0,\boldsymbol{k}} = 0$, $u_{\lambda=0,\boldsymbol{k},n} = c_n^0$, $v_{\lambda=0,\boldsymbol{k},n} = 0$, and $\hat{b}_{\lambda=0,\boldsymbol{k}} = \hat{b}_{\lambda=0,\boldsymbol{k}}^{\dagger} = 1$ [43]. In particular, this convention for mode labeling gives $U_{0\boldsymbol{k},0\boldsymbol{q}} = n_0$, $V_{0\boldsymbol{k},0\boldsymbol{q}} = 0$, $W_{0\boldsymbol{k},0\boldsymbol{q}} = 0$, reproducing Eq. (3).

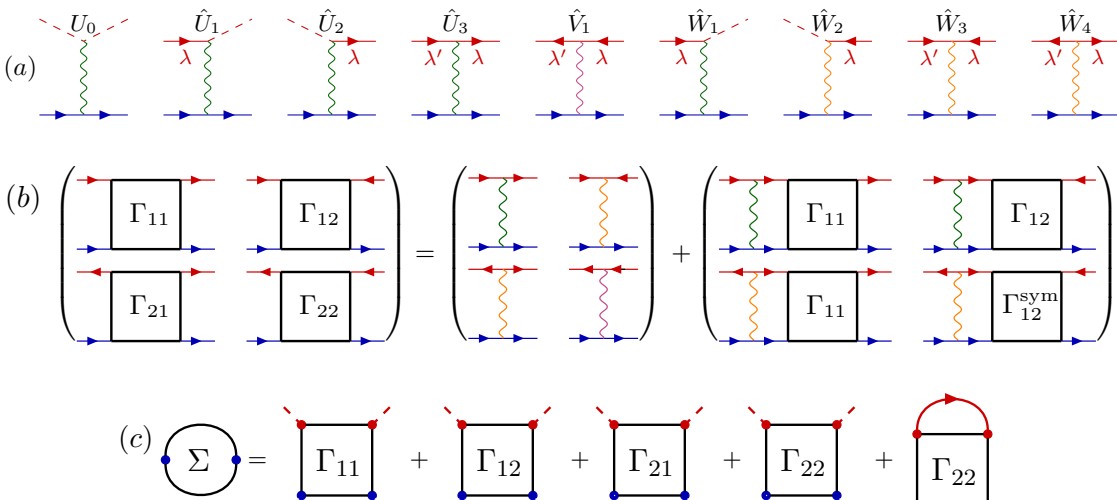

Figure 2: (a) The different impurity-boson interaction processes given by Eqs.(6)-(13). Solid blue lines denote the impurity, dashed red lines without arrows denote the Gutzwiller mean-field ground state of the bath ($\lambda = 0$), and solid red lines with arrows denote elementary excitations of the bath ($\lambda \neq 0$). Wavy lines denote $U$ (green), $V$ (purple), and $W$ (orange). The $\hat{W}_1$ and $\hat{W}_2$ vertices represent two (identical) processes and have been symmetrized, see Eqs. (10) and (11). (b) Bethe-Salpeter equation for the in-medium scattering matrix at zero temperature given by Eq. (14). The different vertices shown in panel (a) lead to four different scattering processes since the number of bosonic excitations is not conserved in general. (c) Generalized ladder approximation for the impurity self-energy at zero temperature.

The interaction in Eq. (4) separates into a mean-field contribution $U_0 = n_0$ and the processes

$$\hat{U}_1 = \sum_{\mathbf{p},\mathbf{q}} \sum_{\lambda>0} U_{00,\lambda\mathbf{p}} \hat{c}_{\mathbf{q}}^\dagger \hat{c}_{\mathbf{q}-\mathbf{p}} \hat{b}_{\mathbf{p},\lambda}, \tag{6}$$

$$\hat{U}_2 = \sum_{\mathbf{p},\mathbf{q}} \sum_{\lambda>0} U_{\lambda\mathbf{p},00} \hat{c}_{\mathbf{q}}^\dagger \hat{c}_{\mathbf{q}+\mathbf{p}} \hat{b}_{\mathbf{p},\lambda}^\dagger, \tag{7}$$

$$\hat{U}_3 = \sum_{\mathbf{p},\mathbf{q},\mathbf{k}} \sum_{\lambda\lambda'>0} U_{\lambda\mathbf{k},\lambda'\mathbf{p}} \hat{c}_{\mathbf{q}}^\dagger \hat{c}_{\mathbf{q}-\mathbf{p}+\mathbf{k}} \hat{b}_{\mathbf{k},\lambda}^\dagger \hat{b}_{\mathbf{p}\lambda'}, \tag{8}$$

$$\hat{V}_1 = \sum_{\mathbf{p},\mathbf{q},\mathbf{k}} \sum_{\lambda\lambda'>0} V_{\lambda\mathbf{k},\lambda'\mathbf{p}} \hat{c}_{\mathbf{q}}^\dagger \hat{c}_{\mathbf{q}+\mathbf{p}-\mathbf{k}} \hat{b}_{\mathbf{k},\lambda} \hat{b}_{\mathbf{p},\lambda'}^\dagger, \tag{9}$$

$$\hat{W}_1 = \frac{(2)}{2} \sum_{\mathbf{p},\mathbf{q}} \sum_{\lambda>0} W_{00,\lambda\mathbf{p}} \hat{c}_{\mathbf{q}}^\dagger \hat{c}_{\mathbf{q}-\mathbf{p}} \hat{b}_{\mathbf{p},\lambda}, \tag{10}$$

$$\hat{W}_2 = \frac{(2)}{2} \sum_{\mathbf{p},\mathbf{q}} \sum_{\lambda>0} W_{\lambda\mathbf{p},00} \hat{c}_{\mathbf{q}}^\dagger \hat{c}_{\mathbf{q}+\mathbf{p}} \hat{b}_{\mathbf{p},\lambda}^\dagger, \tag{11}$$

$$\hat{W}_3 = \frac{1}{2} \sum_{\mathbf{p},\mathbf{q},\mathbf{k}} \sum_{\lambda\lambda'>0} W_{\lambda\mathbf{p},\lambda'\mathbf{k}} \hat{c}_{\mathbf{q}}^\dagger \hat{c}_{\mathbf{q}-\mathbf{p}-\mathbf{k}} \hat{b}_{\mathbf{p},\lambda} \hat{b}_{\mathbf{k}\lambda'}, \tag{12}$$

$$\hat{W}_4 = \frac{1}{2} \sum_{\mathbf{p},\mathbf{q},\mathbf{k}} \sum_{\lambda\lambda'>0} W_{\lambda\mathbf{p},\lambda'\mathbf{k}} \hat{c}_{\mathbf{q}}^\dagger \hat{c}_{\mathbf{q}+\mathbf{p}+\mathbf{k}} \hat{b}_{\mathbf{p},\lambda}^\dagger \hat{b}_{\mathbf{k}\lambda'}^\dagger, \tag{13}$$

illustrated diagrammatically in Fig. 2(a). Here, the factors of 2 in Eqs. (10) and (11) arise from the Bose statistics of the fluctuations and account for the symmetry of the $W$-vertex under dummy label exchange [5]. Consequently, the corresponding processes $\hat{W}_1$ and $\hat{W}_2$ can be represented as in Fig. 2(a) or, equivalently, with the appropriate excitation line on the opposite side of the vertex. As discussed in Refs. [27, 28, 28–30], the vertices must be calculated numerically at each point on the phase diagram, with analytic forms available only in limiting cases, see App. A.

From Fig. 2(a), we see that processes $\hat{U}_1$, $\hat{U}_2$, $\hat{W}_1$, and $\hat{W}_2$ involve scattering between an impurity, the Gutzwiller mean field, and an elementary excitation. Although $\hat{U}_1$ and $\hat{W}_1$ describe the same process, albeit with different interaction strengths, we distinguish them, as well as $\hat{U}_2$ and $\hat{W}_2$, due to distinctness of the underlying physics. The remaining processes involve the impurity and elementary excitations of the bath, allowing for the involvement of different modes within a single scattering process. This multimodal structure adds another layer of complexity compared to the usual Bogoliubov theory, which describes only processes between the impurity, Bogoliubov phonons, and the condensate. This approach is unsuitable for describing the quantum critical region and the Mott insulator phase where condensate depletion ranges from significant to total and many modes become important [42].

The rules for constructing Feynman diagrams follow the well-known formalism for bosons [5], with fluctuation operators satisfying bosonic commutation relations $[\hat{b}_{\lambda,\mathbf{k}}, \hat{b}_{\lambda',\mathbf{k}}^\dagger] = \delta_{\mathbf{k},\mathbf{k}'}\delta_{\lambda,\lambda'}$. Explicitly, a solid red line corresponds to a Green's function $D_{\lambda>0}^{(0)}(\mathbf{q},z) = 1/(z-\omega_{\mathbf{q},\lambda})$ for a bosonic bath excitation, while a solid blue line corresponds to a noninteracting (bare) impurity Green's function $G^{(0)}(\mathbf{q},z) = 1/(z-\varepsilon_{\mathbf{q}})$. Dashed red lines without arrows correspond to the $\lambda = 0$ mode, i.e. interaction with the mean-field Gutzwiller ground state. Here, $z$ denotes a Matsubara frequency, and we perform the usual analytical continuation $z \to \omega + i0_+$ at the end of the calculation to obtain the retarded impurity Green's function.

The processes $\hat{W}_3$ or $\hat{W}_4$ involve the annihilation and creation of two modes, respectively, and diagrams describing both permutations of the outputs contribute and must be

summed over. Additionally, at zero temperature, any diagrams that involve internal back-propagating excitation lines make vanishing contributions due to zero-mode occupation $n_{\lambda,\mathbf{k}} = \langle \hat{b}_{\lambda,\mathbf{k}}^{\dagger} \hat{b}_{\lambda,\mathbf{k}} \rangle = 0$.

### 3.1.2 Impurity-boson scattering

Having established a general diagrammatic framework for analyzing the properties of a mobile impurity in a BH bath, we now develop an approximate scheme for solving this in the regime of strong impurity-boson interactions. In Ref. [27], this problem was studied in the perturbative regime with impurity properties evaluated to quadratic order $(U_{\mathrm{IB}}/U)^2$ (see App. A.3). Here, we consider a range of coupling strengths beyond even $|U_{\mathrm{IB}}/U| \sim \mathcal{O}(1)$ where truncation in powers of the impurity-bath coupling is no longer justified, and selected classes of higher-order Feynman diagrams must be included. Our approach is inspired by the remarkable accuracy of the ladder approximation for describing impurities in a Fermi gas, i.e. the Fermi polaron [7], as well as its good agreement with experimental data when describing impurities in a BEC [9]. We therefore expect a ladder approximation suitably generalized to the case at hand to be reliable even at strong interactions, since a strongly repulsive Bose gas in a qualitative sense is in between a weakly interacting BEC and a Fermi gas. As we shall see, this expectation is confirmed to a remarkable degree by comparing with Monte-Carlo calculations.

We begin by deriving a set of coupled equations for the in-medium scattering matrices $\Gamma_{ij}^{\lambda\lambda'}$. In analogy with Beliaev theory for a weakly interacting BEC [5], the scattering is described by a matrix in which the different matrix elements correspond to the incoming and outgoing excitations whose number is not conserved. Resumming scattering events with the bare vertices shown in Fig. 2(a) within a ladder approximation generalized to the present case yields the coupled Bethe-Salpeter equations

$$
\begin{aligned}
\Gamma_{11}^{\lambda\lambda'}(P,\mathbf{p},\mathbf{p}') &= U_{\mathrm{IB}}U_{\lambda\mathbf{p},\lambda'\mathbf{p}'} + \frac{U_{\mathrm{IB}}}{\sqrt{M}}\sum_{\mathbf{p}_1,\lambda_1>0}\frac{U_{\lambda\mathbf{p},\lambda_1\mathbf{p}_1}\Gamma_{11}^{\lambda_1\lambda'}(P,\mathbf{p}_1,\mathbf{p}')}{z-\omega_{\mathbf{p}_1,\lambda_1}-\varepsilon_{\mathbf{P}-\mathbf{p}_1}},\\[4pt]
\Gamma_{12}^{\lambda\lambda'}(P,\mathbf{p},\mathbf{p}') &= U_{\mathrm{IB}}\tilde{W}_{\lambda\mathbf{p},\lambda'\mathbf{p}'} + \frac{U_{\mathrm{IB}}}{\sqrt{M}}\sum_{\mathbf{p}_1,\lambda_1>0}\frac{U_{\lambda\mathbf{p},\lambda_1\mathbf{p}_1}\Gamma_{12}^{\lambda_1\lambda'}(P,\mathbf{p}_1,\mathbf{p}')}{z-\omega_{\mathbf{p}_1,\lambda_1}-\varepsilon_{\mathbf{P}-\mathbf{p}_1}},\\[4pt]
\Gamma_{21}^{\lambda\lambda'}(P,\mathbf{p},\mathbf{p}') &= U_{\mathrm{IB}}\tilde{W}_{\lambda\mathbf{p},\lambda'\mathbf{p}'} + \frac{U_{\mathrm{IB}}}{\sqrt{M}}\sum_{\mathbf{p}_1,\lambda_1>0}\frac{\tilde{W}_{\lambda\mathbf{p},\lambda_1\mathbf{p}_1}\Gamma_{11}^{\lambda_1\lambda'}(P,\mathbf{p}_1,\mathbf{p}')}{z-\omega_{\mathbf{p}_1,\lambda_1}-\varepsilon_{\mathbf{P}-\mathbf{p}_1}},\\[4pt]
\Gamma_{22}^{\lambda\lambda'}(P,\mathbf{p},\mathbf{p}') &= U_{\mathrm{IB}}V_{\lambda\mathbf{p},\lambda'\mathbf{p}'} + \frac{U_{\mathrm{IB}}}{\sqrt{M}}\sum_{\mathbf{p}_1,\lambda_1>0}\frac{\tilde{W}_{\lambda\mathbf{p},\lambda_1\mathbf{p}_1}\hat{\mathcal{S}}[\Gamma_{12}^{\lambda_1\lambda'}(P,\mathbf{p}_1,\mathbf{p}')]}{z-\omega_{\mathbf{p}_1,\lambda_1}-\varepsilon_{\mathbf{P}-\mathbf{p}_1}},
\end{aligned}
\tag{14}
$$

where we have introduced the shorthand notation $P \equiv (\mathbf{P}, z)$ for the center-of-mass momentum/energy. Equation (14) is shown diagrammatically in Fig. 2(b), which illustrates that $\Gamma_{11}^{\lambda\lambda'}(P,\mathbf{p},\mathbf{p}')$ describes the scattering of the impurity on an excitation of the bath, $\Gamma_{12}^{\lambda\lambda'}(P,\mathbf{p},\mathbf{p}')$ and $\Gamma_{21}^{\lambda\lambda'}(P,\mathbf{p},\mathbf{p}')$ describe the impurity annihilating/emitting two excitations, and $\Gamma_{22}^{\lambda\lambda'}(P,\mathbf{p},\mathbf{p}')$ describe the impurity scattering on a "hole" excitation, all with in- and out-going relative momenta $\mathbf{p}$ and $\mathbf{p}'$. These processes are coupled due to the interactions in Eqs. (6)-(13). We have used the shorthand notation $\tilde{W}_{\lambda\mathbf{p},\lambda'\mathbf{p}'} = (1 - \delta_{\lambda,0}\delta_{\lambda',0}/2)W_{\lambda\mathbf{p},\lambda'\mathbf{p}'}$ in order to compress four distinct Bethe-Salpeter equations describing the possible scatterings between the impurity, Gutzwiller mean-field ground state, and elementary excitations into one expression in Eq. (14) that applies for all $\lambda$, $\lambda'$. Also, one must symmetrize $\Gamma_{12}^{\lambda\lambda'}$ and $\Gamma_{21}^{\lambda\lambda'}$ for $\lambda, \lambda' > 0$ due to the indistinguishability of the two incoming or outgoing elementary excitations, respectively, which produces a factor of 2 when they are integrated over internally in a diagram.

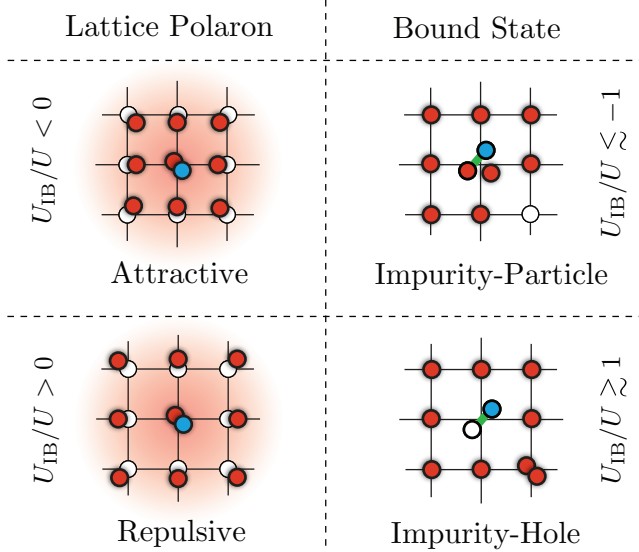

Figure 3: For $U_{\text{IB}} < 0$, the impurity (blue) attracts the surrounding bosons (red) forming an attractive polaron. For $U_{\text{IB}}/U \lesssim -1$, it forms a bound state with one extra boson. For $U_{\text{IB}} > 0$, the impurity repels the surrounding bosons forming a repulsive polaron. For $U_{\text{IB}}/U \gtrsim 1$, it pushes away the bosons completely forming a bound state with a hole.

This is taken into account in the final line of Eq. (14) by the symmetrizer $\hat{\mathcal{S}}$ which acts as $1 + \hat{P}$ when the in-medium scattering matrix element involves two elementary excitations with the transposition operator $\hat{P}$. When $\lambda = 0$ is involved, no symmetrization is performed ($\hat{\mathcal{S}} = 1$) due to distinguishability. Finally, the Gutzwiller mean-field ground state is excluded from the summations in Eq. (14) as it would give rise to disconnected diagrams.

Physically, the Bethe-Salpeter equation is an in-medium generalization of the Lippman-Schwinger equation for the scattering of two particles in a vacuum. The denominators in the summations in Eq. (14) describe the propagation of a bare impurity and a bath excitation and their poles give rise to a scattering continuum, which is bounded from above due to the lattice. For example, $\varepsilon_{\mathbf{k}}$ has a width of $8t$, while the Goldstone mode in the Bogliubov theory of the bath has a width $\sqrt{(8t)^2 + 8t|\psi_0|^2 U}$, decreasing from the deep superfluid to the quantum critical regime. Note that in our calculations, finite size effects turn this continuum into a discrete set of states which can be important for small systems; see App. A.4.

In the vacuum limit of a single boson and impurity, the in-medium scattering matrix reduces to the two-body scattering matrix with discrete poles at the energies of any bound states consisting of one boson and one impurity. The ladder approximation therefore includes strong two-body impurity-boson correlations and bound states exactly in a many-body environment. This is crucial as we shall see, since the presence of several different impurity-boson dimer states, illustrated in Fig. 3, significantly influences the properties of the impurity, leading to the existence of several polarons, in analogy with what was found in the simpler case when the bosons are deep in the superfluid regime [42].

In practice, at zero temperature, the in-medium scattering matrix can be solved by first obtaining elements $\Gamma_{11}$ and $\Gamma_{12}$ self-consistently. The remaining components can be obtained directly through matrix multiplication. Details of this numerical procedure are described in App. A.6.

### 3.1.3 Impurity self-energy

The properties of the impurity interacting with the lattice bosons are described within our approach by the interacting impurity Green's function

$$G(\mathbf{q}, z) = \frac{1}{z - \varepsilon_{\mathbf{q}} - \Sigma(\mathbf{q}, \omega)} \, , \tag{15}$$

where $\Sigma(\mathbf{q}, z)$ is the impurity self-energy. In particular, the energy $E_{\mathbf{k}}$ of any quasiparticle, which we denote as a polaron in the present context, is determined by solving $E_{\mathbf{k}} = \varepsilon_{\mathbf{k}} + \text{Re}\Sigma(\mathbf{k}, E_{\mathbf{k}})$ [4].

Having derived a Bethe-Salpeter expression for the scattering between the impurity and the bosons, we can now build this into a self-energy for the impurity. Figure 2(c) shows our generalized ladder approximation for the impurity self-energy, which contains two classes of diagrams. The first four diagrams in Fig. 2(c) correspond to the impurity scattering particles out of the mean-field ground state of the bath ($\lambda = 0$), i.e. $\Gamma_{ij}^{00}$. In particular, $\Gamma_{11}^{00}$ includes the mean-field energy $U_{\text{IB}} n_0$. Unlike the Bogoliubov approach, where the expansion is based on the condensate density considering a condensate fraction close to unity, our method is based on expanding the total density and therefore remains valid even near the critical point. Consequently, the usual product of the condensate density and the in-medium scattering matrix (c.f. [42, 44]) does not appear explicitly in our QGW method and is only recovered in the weakly interacting BEC limit. This is due to the broader applicability of the QGW approach, in particular to the quantum critical regime where the condensate is largely or totally depleted.

The last diagram in Fig. 2(c) involves a loop sum over collective modes, and at zero temperature this process is only non-zero for the $\Gamma_{22}^{\lambda\lambda}$ in-medium scattering matrix element, contributing for all $\lambda > 0$. We note that this involves processes not taken into account in the usual diagrammatic expansion method based on the Fröhlich model [45], which was shown in Ref. [27] to provide a poor description of the physics particularly in the Mott and quantum critical regimes. It is also insufficient to describe the Bose polaron deep in the superfluid regime [44, 46]. Furthermore, this diagram contains $U_{\text{IB}} \langle \delta_2 \hat{n}(\mathbf{r}) \rangle$, the quantum correction to the mean-field coupling due to quantum fluctuations [27–29]. Explicit expressions for the self-energy diagrams can be found in App. A.3 for the interested reader. Finally, we note that when iterating the in-medium scattering matrix to second order, the self-energy in Fig. 2(c) recovers the first and second order terms considered in Ref. [27].

### 3.1.4 Self-consistency

So far, the internal impurity Green's functions in our diagrams have been non-interacting (bare), which corresponds to assuming a non-interacting impurity in the scattering processes. Technically, this can be seen from the fact that the bare impurity energy $\varepsilon_{\mathbf{p}}$ appears in the denominators of Eq. (14), which therefore describe the pair propagation of a non-interacting impurity and a bath excitation. In the first four diagrams of Fig. 2(c), this in turn gives rise to a scattering continuum of states consisting of a non-interacting impurity and one bath excitation with energies spanning $\min_{\mathbf{p}}(\omega_{\mathbf{p},\lambda} + \varepsilon_{\mathbf{P}-\mathbf{p}}) \leq \omega \leq \max_{\mathbf{p}}(\omega_{\mathbf{p},\lambda} + \varepsilon_{\mathbf{P}-\mathbf{p}})$ with $\mathbf{P}$ the total momentum. Likewise, in the last diagram of Fig. 2(c), it gives rise to a continuum of states of a non-interacting impurity and two bath excitations with energies $\min_{\mathbf{p},\mathbf{q}}(\omega_{\mathbf{p},\lambda} + \omega_{\mathbf{q},\lambda'} + \varepsilon_{\mathbf{P}-\mathbf{p}-\mathbf{q}}) \leq \omega \leq \max_{\mathbf{p},\mathbf{q}}(\omega_{\mathbf{p},\lambda} + \omega_{\mathbf{q},\lambda'} + \varepsilon_{\mathbf{P}-\mathbf{p}-\mathbf{q}})$. This is unphysical since the impurity in the intermediate scattering states of course interacts with its environment.

Since we are interested in the interplay between these scattering continua and polaron formation, we now address this problem.

In principle, the problem can be solved by using full interacting impurity Green's functions everywhere in the diagrams, and such a self-consistent approximation has indeed been implemented for an impurity in a weakly interacting BEC [44], as well as for two-component Fermi mixtures [47–49].

The self-consistent approximation can be implemented using an iterative procedure via successive replacements $\varepsilon_{\mathbf{q}} \rightarrow \varepsilon_{\mathbf{q}} + \Sigma(\mathbf{q}, \omega)$ in the calculation of the in-medium scattering matrix. In practice, we find that the numerical cost even of the order $\mathcal{O}(10^1)$ iterations (as done in Ref. [44]) becomes rapidly prohibitive as the calculation of the vertices and associated Bethe-Salpeter equation is fully numerical in the quantum critical regime. Issues that arise through the numerical implementation of the self-consistent approximation already at the single iteration level are discussed further in App. A.7 for the interested reader. We find instead that the simple replacement $\varepsilon_{\mathbf{q}} \rightarrow \varepsilon_{\mathbf{q}} + U_{\text{IB}} \langle \hat{n} \rangle$ in the Bethe-Salpeter equation, i. e. adding the mean-field shift to the impurity energies in the intermediate states, is sufficient to capture the main effects of self-consistency over the range of couplings considered in this work. Note that since there is only one impurity, we assume that the bath is essentially unaffected in the thermodynamic limit so that we can use bare bath Green's functions.

## 3.2 Full configuration interaction quantum Monte Carlo (QMC)

In this section, we discuss a completely different method to analyze an impurity immersed in a strongly correlated boson gas, which in principle allows for an exact calculation of the ground state properties. Full configuration interaction QMC is a type of projector Monte Carlo, which can be used to stochastically sample the ground state eigenpair $(E_0, \mathbf{v}_0)$ of a matrix $\mathbf{H}$ [50]. In the context of this paper, $\mathbf{H}$ is a matrix realization of the Hamiltonian defined in Eq. (1), $E_0$ is its ground state energy, and $\mathbf{v}_0$ a vector representation of the ground state. Full configuration interaction QMC samples these quantities by repeatedly stochastically applying the following scheme to an arbitrarily chosen initial coefficient vector $\mathbf{c}^{(0)}$:

$$\mathbf{c}^{(n+1)} = \mathbf{c}^{(n)} + \delta\tau \left( S^{(n)} \mathbf{1} - \mathbf{H} \right) \mathbf{c}^{(n)}, \tag{16}$$

where, $\delta\tau$ is a (small) time step, $\mathbf{1}$ the identity matrix and $S^{(n)}$ a scalar energy shift that is updated after every step in order to keep the 1-norm of the vectors $\mathbf{c}^{(n)}$ constant [51]. For a large number of steps $n$, the iteration converges such that the expectation value of the shift $S^{(n)}$ coincides with the ground state energy $E_0$ and the vectors $\mathbf{c}^{(n)}$ fluctuate around the eigenvector $\mathbf{v}_0$. In practice, the ground state energy is estimated from a sample mean of $S^{(n)}$ excluding steps from an initial equilibration phase. See App. B for detailed information on the method.

While the Bose-Hubbard Hamiltonian does not exhibit the QMC sign problem, and the method itself is in principle exact, the noise in the shift introduces a statistical bias, also known as the population control bias [52–54]. We apply importance sampling [55–57] as a similarity transformation to the Hamiltonian matrix based on an optimized guiding vector that approximates the exact ground state. Importance sampling brings the dual benefit of reducing the sampling noise and with it the stochastic uncertainty in the energy estimators as well as suppressing the population control bias to undetectable levels. Importance sampling was essential to obtain high quality results for the largest system sizes reported in this work (100 bosons and 1 impurity particle in a $10 \times 10$ lattice with a Hilbert space dimension of $\approx 10^{60}$), where without importance sampling, the variance of the energy would be so large to be effectively unusable. More details on the importance sampling and optimization of the guiding vector can be found in App. B.2. All QMC results reported in this work were obtained with the open-source library `Rimu.jl` [58].

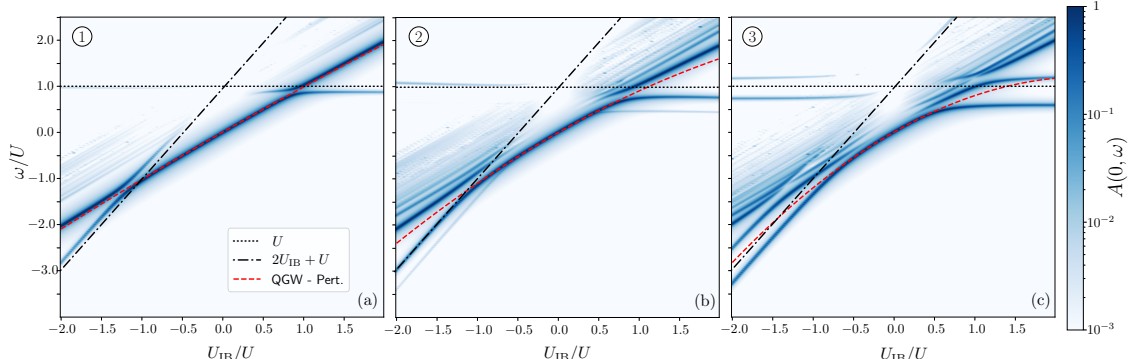

Figure 4: Impurity spectral function $A(\mathbf{k} = 0, \omega)$ for the three points ①, ② and ③ in the phase diagram in Fig. 1. (a) The Mott insulator phase with $\mu/U = \sqrt{2} - 1$, $4t/U = 0.1292$. (b) The $O(2)$ point (coming from the superfluid side) with $\mu/U = 0.4142$, $4t/U = 0.1723$ and condensate fraction 0.01. (c) The superfluid phase with $\mu/U = 0.4014$, $4t/U = 0.2154$ and condensate fraction 0.3. The dashed red lines are the energy of the lattice polaron within second-order perturbation theory [27], the black lines are the energy $U$ for strong repulsion, and the black dash-dotted lines are the energy $2U_{IB} + U$ for strong attraction.

## 4 Results

We now present numerical results for the impurity properties obtained from our QGW and QMC calculations.[2] We focus on the vicinity of the $O(2)$ point of the MI-SF transition taking the filling fraction to be at or very close to unity. The filling factor is calculated by choosing the value of $\mu/U$ such that $\langle \hat{n} \rangle = n_0 + \langle \delta_2 \hat{n} \rangle$ is held fixed (Eq. (A.9) in App. A). The correction $\langle \delta_2 \hat{n} \rangle$ accounts for zero point quantum fluctuations of the bath which modify the filling particularly in the quantum critical regime, while it is zero in the Mott phase. We focus on the case of zero momentum, leaving momentum-dependent lattice polaron properties, such as the full quasiparticle dispersion and effective mass, as the subject of future work.

In Secs. 4.1.1-4.1.3, we analyze the impurity properties obtained using our diagrammatic approach as a function of the impurity-boson interaction strength at the fixed points indicated in the phase diagram in Fig. 1(a). In Sec. 4.2-4.3, we consider the impurity properties across the $O(2)$ MI-SF phase transition for fixed impurity-boson interaction strength. Finally, Sec. 4.3 compares the QGW results with those of the QMC calculations.

### 4.1 Results for a fixed bath and variation of $U_{IB}$

#### 4.1.1 Mott insulating bath

We first consider the case where the bosons are in the MI phase taking $4t/U = 0.1292$ corresponding to the point labeled ① in Fig. 1. In this regime, the Gutzwiller mean-field ground state is an incompressible Fock state, and the only allowed excitations in the bath are gapped particle and hole excitations, which must occur in pairs due to number conservation. The individual gaps of the particle and hole excitations depend on the chemical potential of the bath [59], whereas the energy gap to excite a particle-hole pair does not.

In Fig. 4 (a), QGW results for the impurity spectral function $A(\mathbf{p}, \omega) = -2\text{Im}G(\mathbf{p}, \omega)$ are shown as a function of the impurity-boson interaction strength. In the MI phase, the first

---

[2]The QGW results presented in this work were performed with finite cutoffs for modes ($\lambda \leq 2$) and occupation numbers ($n \leq 7$) and checked for convergence by performing calculations for increased cutoffs.

diagram in Fig. 2(c) contributes to the self-energy with a mean field term, the three middle diagrams are zero due to particle conservation, whereas the last term contributes by dressing the impurity with bosons depleted from the mean-field ground state due to boson-boson interactions. Such particle-hole scatterings are possible even at zero temperature via the process $\hat{V}_1$, which contributes quantum corrections to the filling factor. Additionally, at order $\mathcal{O}(U_{\text{IB}}^2)$, the $\hat{W}_3$ and $\hat{W}_4$ processes can describe the virtual excitation and de-excitation of a particle-hole pair in the polaron cloud [27]. In our ladder approximation, such processes are generalized to strong interactions by inserting an infinite number of re-scatterings of the impurity and an excitation (hole or particle) via the process $\hat{U}_3$.

We see that the impurity spectral function in Fig. 4(a) exhibits sharp lines corresponding to quasiparticles, i.e. polarons. For weak impurity-boson interaction strength, the ground state of the system is a well-defined polaron with a large spectral weight and an energy close to the mean-field value $U_{\text{IB}}\langle \hat{n} \rangle$. This extends over a wide range of interaction strengths $|U_{\text{IB}}/U| \lesssim 1$, reflecting the incompressible nature of the MI phase, which is relatively insensitive to the impurity. The closely lying faint lines above the polaron line in Fig. 4(a) are a finite-size version of the scattering continuum in the thermodynamic limit consisting of the impurity and two bath (particle-hole) excitations as discussed in Sec. 3.1.4, which is separated in energy from the polaron ground state by the Mott gap; see Fig. 1(b). These finite-size effects appear because we use a lattice with $10 \times 10$ sites in our diagrammatic calculations, as detailed in App. A.4.

Figure 4(a) also shows a pair of avoided crossings at $|U_{\text{IB}}/U| \simeq 1$. Here, the ground state changes abruptly as a non-perturbative regime emerges for stronger interactions. For attractive interaction $U_{\text{IB}}/U \lesssim -1$, the attractive polaron energy depends linearly on $U_{\text{IB}}$ with a larger slope than in the mean-field regime. This can be understood from the ability of the impurity to bind to one of the bosons to form a dimer state, as illustrated in Fig. 3. The energy of such a dimer state is approximately $2U_{\text{IB}} + U$ since two bosons occupy the same site as the impurity. This agrees well with the obtained energy of the attractive polaron, see Fig. 4 (a)

The horizontal line in Fig. 4(a) for a strong repulsive impurity-boson interaction $U_{\text{IB}}/U \gtrsim 1$ corresponds to a repulsive polaron with an energy independent of $U_{\text{IB}}$, which can also be interpreted in terms of bound-state physics. In this case, the impurity pushes bosons away from the site it occupies, which is equivalent to an "excitonic" bound state between the impurity and a hole as illustrated in Fig. 3. When the impurity pushes away a boson, it creates a site containing two bosons, and the energy of this repulsive polaron is therefore $U$, which agrees well with the numerical results.

The particle-hole symmetry at unit filling discussed in Sec. 3.1 can in fact be used to show that the spectrum is symmetric with respect to $U_{\text{IB}} \longleftrightarrow -U_{\text{IB}}$ in the vicinity of the $O(2)$ point aside from the mean-field shift. This symmetry is recovered by our diagrammatic theory. From a technical point of view, these bound states give rise to poles in the in-medium scattering matrix entering the impurity self-energy, which are related by a particle-hole symmetry at unit filling. Indeed, since the process $\hat{U}_3$ entering as an infinite series inside the last diagram of Fig. 2(c) is of equal magnitude but opposite sign for impurity-particle and impurity-hole processes at unit filling (see App. A.2), the poles of the in-medium scattering matrix are invariant with respect to the sign of $U_{\text{IB}}/U$ aside from the mean-field shift $U_{\text{IB}}\langle \hat{n} \rangle$ thus giving the above relation between the attractive and repulsive polaron energies.

Interestingly, Fig. 4 (a) shows a strong well-defined polaron branch above the ground state in the strong interaction regime. Since its energy closely follows the mean-field value $U_{\text{IB}}\langle \hat{n} \rangle$, we denote this branch as a "mean-field" polaron, which physically corresponds to the impurity moving in an incompressible background. This mean-field polaron is well defined because it is separated in energy from the continuum of states because of the excitation gap of the MI. Finally, there is also a faint horizontal line in Fig. 4(a) close to the energy $\sim U$ for

strong attraction. This is a continuation of the repulsive polaron state in the regime of strong attractions, where it becomes an "upper polaron" as an excited state stabilized by the fact that it is above the scattering continuum.

### 4.1.2 Critical bath – $O(2)$ point

We now turn our attention to the intriguing and challenging region in the vicinity of the $O(2)$ transition where the bosonic bath is highly correlated. This transition occurs for $\mu/U = \sqrt{2}-1$ and $4t/U = (\sqrt{2}-1)^2$ in the mean-field Gutzwiller calculation [59].

As the $O(2)$ point is approached from the MI phase, the gap of individual particle or hole excitations closes, see Fig. 1(b), and the elementary excitations become pure phase (Goldstone) and amplitude (Higgs) modes. Both modes are gapless due, respectively, to the spontaneous breaking of $U(1)$ phase symmetry and Lorentz invariance at the $O(2)$ point in the low-energy effective field theory [60]. The Higgs mode is gapless only at the transition point, as is typical for a quantum phase transition.

In Fig. 4 (b), we show the impurity spectral function as a function of $U_{\text{IB}}$ but now for $4t/U = 0.1723$, corresponding to the point labeled ② in Fig. 1 just on the superfluid side of the $O(2)$ point. Here, the ground state at unit filling is accurately described by a superposition of the lowest number states $|0\rangle$, $|1\rangle$, and $|2\rangle$ at each lattice site [35], where a large amplitude of the $|1\rangle$ state corresponds to an insulator-like component and small and equal amplitudes for the $|0\rangle$ and $|2\rangle$ states dictate the properties of the particle-hole symmetric superfluid component. The compressibility of the bath increases due to the condensate and the sound speed takes a finite value [59]. Since a non-zero superfluid density in our approach corresponds to the breaking of particle number conservation, the processes $\hat{U}_1$, $\hat{U}_2$, $\hat{W}_1$, and $\hat{W}_2$ given by Eqs. (6), (7), (10) and (11) now contribute with $\hat{U}_3$ re-scatterings, see Fig. 2(a) and App A.2. Consequently, all elements of the in-medium scattering matrix are non-zero and all diagrams in Fig. 2(c) now contribute to the self-energy.

First, we see from from Fig. 4 (b) that in the weak interaction regime, there is still a well-defined polaron with an energy close to the mean-field value, as expected. However, because of the larger compressibility of the bath, the mean-field region is smaller than in the MI phase, and second-order effects are stronger. Also, the scattering continuum now appears just above the mean-field energy since the excitations of the bath are gapless, see Fig. 1(b). Perturbation theory breaks down already at $|U_{\text{IB}}|/U \simeq 0.5$, and we see that *two* kinds of polarons emerge below the continuum as the strongly interacting regime is entered. The polarons with the largest spectral weight for $U_{\text{IB}}/U \gtrsim 0.5$ and $U_{\text{IB}}/U \lesssim -0.5$ are smoothly connected to the repulsive and attractive polarons in the MI discussed above with almost the same energy. This is because the character of the Mott insulating ground state is inherited across the quantum phase transition into the superfluid phase, a phenomenon which is commonly referred to as "Mottness" [34].

Interestingly, Fig. 4(b) shows that a new polaron with a smaller spectral weight and lower energy emerges for strong attractive and repulsive interactions. The origin of this new polaron branch, which is now the ground state, is a strong dressing of the impurity by the gapless modes of the bosonic bath, which significantly lowers its energy. This dressing comes from the first four diagrams in Fig. 2(c), which are zero in the MI phase. Because of the small condensate fraction, the residue of this new attractive and repulsive polaron is very small, and hence the spectral line is faint. The appearance of a new polaron branch is due to the non-analytic behavior of the excitation spectrum of the bosonic bath, with both the Higgs and the Goldstone modes becoming gapless at the $O(2)$ transition point. This will be analyzed further in Sec. 4.2. Finally, we see that the upper polaron with energy $\sim U$ in the strongly attractive region discussed above for the MI phase remains in the superfluid phase.

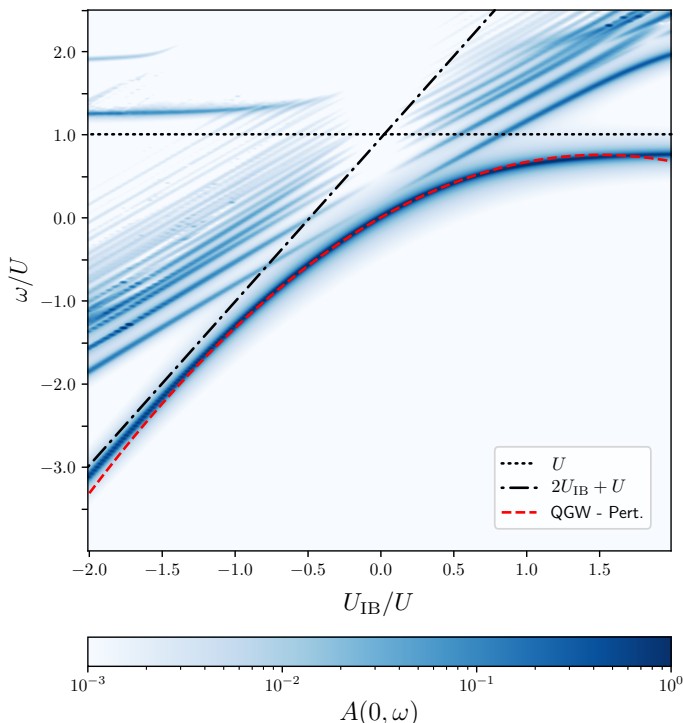

Figure 5: Impurity spectral function $A(\mathbf{k} = 0, \omega)$ at unit filling line deeper into the superfluid phase with $\mu/U = 0.3021$, $4t/U = 0.4$, and condensate fraction 0.74. Lines are as in Fig. 4.

### 4.1.3 Superfluid bath

We now explore the spectral properties of the impurity deeper in the superfluid phase at unit filling taking $4t/U = 0.2154$ corresponding to the point labeled ③ in Fig. 1(a). The corresponding QGW results for the impurity spectral function as a function of the interaction strength $U_{IB}$ are shown in Fig. 4(c).

First, we note that the weakly interacting regime, where the energy of the polaron ground state is given by second-order perturbation theory, has decreased further with the quadratic dependence more prominent, which again reflects the increased compressibility of the bosonic bath.

While the Higgs mode becomes increasingly gapped in the superfluid phase, the Goldstone (Bogoliubov sound) mode remains gapless, and the scattering continuum consequently appears just above the mean-field energy.

For strong repulsive and attractive impurity-boson interactions, the ground state repulsive and attractive polaron, visible as faint lines in Fig. 4(b), have now increased their spectral weight significantly, due to the increased condensate fraction, which makes the Goldstone mode dominating the dressing of the impurity.[3]

As $t/U$ increases further, boson-boson correlations decrease and the bath becomes a weakly interacting BEC. It has been shown that in this regime the QGW approach coincides with the Bogoliubov theory for the BH model [28, 59]. Therefore, the ground state polarons evolve into the usual repulsive ($U_{IB} > 0$) and attractive ($U_{IB} < 0$) polarons in a weakly interacting lattice BEC studied in Ref. [42]. The polaron branches above the ground state (Fig. 4(c)), remnants of the attractive and repulsive polaron in the MI phase, on the other hand, decrease

---

[3]Formally, this means the final self-energy diagram in Fig. 2(c) decreases since the Higgs mode becomes increasingly gapped, whereas the first four diagrams increase.

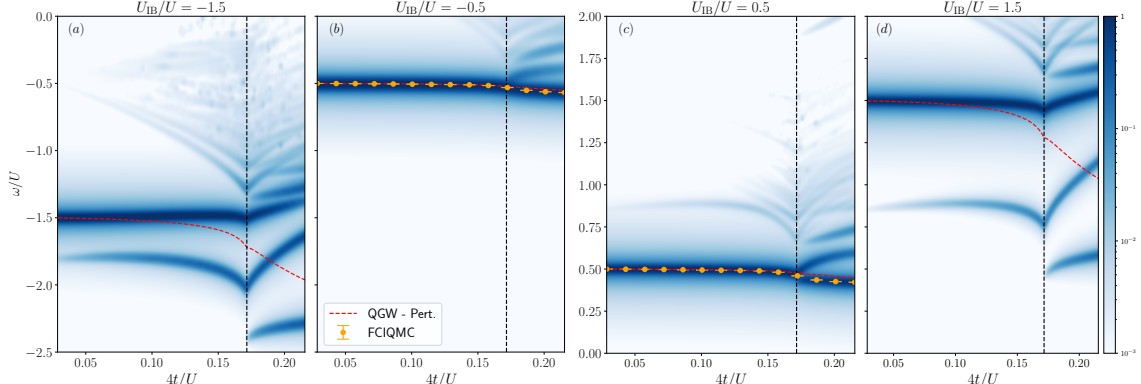

Figure 6: Impurity spectral function $A(\mathbf{k} = 0, \omega)$ across the Mott insulator to super-fluid transition for fixed unit filling $\langle \hat{n} \rangle = 1$ and different impurity-boson interaction strengths. The dashed red lines are the polaron energy obtained from second-order perturbation theory [27]. The dotted black line indicates the $O(2)$ critical point, which occurs at $\mu/U = \sqrt{2} - 1$, $4t/U = (\sqrt{2} - 1)^2$ within mean-field Gutzwiller theory. The QMC results in panel (b) and (c)(yellow dots) are extrapolated from the results for $M = (5 \times 5, 6 \times 6, \dots, 10 \times 10)$ bosons on a square lattice of $M = N^2$ sites and are discussed in Sec. 4.3. The QMC hopping parameter has been rescaled by a factor $t_c/t_{QMC} = 0.7179$ to align the $O(2)$ critical points of the two methods.

in spectral weight with $t/U$. They eventually move into the scattering continuum as shown in Fig. 5 where they become damped as the gap of the Higgs mode increases.

In addition, two kinds of polarons are now visible above the scattering continuum for attractive interactions in Fig. 4(c). Going deeper into the superfluid phase, we find that the polaron with the lowest energy increases its spectral weight, while the one with higher energy, which evolved smoothly from the one in the MI phase, disappears; see Fig. 5. There is also a polaron line above the scattering continuum for strong repulsive interactions. This corresponds to the upper polaron found in the limit of a weakly interacting BEC [42]. It arises from a repulsively bound state consisting of the impurity and a boson, which has been observed experimentally [61].

## 4.2 Results for fixed $U_{IB}$ and variation of the bath

In Fig. 6, we show the polaron spectral function for a few values of $U_{IB}$, across the unit filling MI-SF phase transition by varying $t/U$.[4] The yellow dots in Fig. 6 (b) and (c) represents QMC results that will be discussed in the context of finite-sized systems in the next Sec. 4.3.

Let us first focus on the results for an attractive impurity-boson interaction shown in Fig. 6(a)-(b). The simplest case is for a weak impurity-boson attraction, $U_{IB}/U = -0.5$ shown in panel (b). Here, the polaron energy is fairly constant across the transition and given accurately by perturbation theory. The decrease in polaron energy is due to the compressible nature of the superfluid phase [27]. One can also see how the polaron spectral function clearly inherits the gap closing that characterizes the $O(2)$ critical point of the bath.

---

[4]We note two shortcomings of the QGW model that appear as the limit $U/t \to 0$ is taken in the superfluid phase. First, the Gutzwiller ground state ansatz chosen in Sec. 3.1 describes a condensate with infinite off-diagonal long-range order. In two dimensions, quantum correlations (such as $\langle \delta_2 \hat{n} \rangle$) exhibit infrared divergences according to the Mermin-Wagner theorem. Second, the interaction strengths develop density dependence in the weakly interacting limit not accounted for in the Hamiltonian Eq. (1) [62]. In practice, we have found in previous works [27,29] that these issues do not influence our calculations in the vicinity of the phase transition, but worsen the description as the limit $U/t \to 0$ of the weakly interacting BEC is taken.

For strong impurity-boson attraction $U_{\text{IB}}/U = -1.5$, the situation is very different. In Fig. 6(a), we see two sharp lines in the MI phase clearly separated from the continuum: The ground state attractive polaron and the mean-field polaron as discussed in Sec. 4.1.1. As the gap to the scattering continuum decreases with increasing $t/U$, the attractive polaron energy decreases due to increased dressing. At the phase-transition point, its energy exhibits a cusp, after which it starts to increase in the superfluid phase. In addition, a new attractive polaron appears with a lower energy at the transition point, which arises from the dressing of gapless modes, see Secs. 4.1.2-4.1.3. In particular, Fig. 6(a) shows in a more dramatic way the already discussed failure of perturbation theory in describing the ground state polaron energy in the strong coupling regime, since it cannot describe the formation of impurity-bath bound states.

Due to particle-hole symmetry, we find analogous results across the phase transition for repulsive couplings, as shown in Fig. 6 (c)-(d). Here, we find that the repulsive polaron energies are independent of $U_{\text{IB}}$ for strong interactions due to the association with the impurity-hole (exciton) bound state as discussed previously in Sec. 4.1.1.

Our results promote the intriguing prospect of using polaron spectroscopy in the quantum critical regime for quantum sensing. We note that such non-analytic features appear in many observables at the transition point, including the magnitude of zero-point quantum fluctuations and the gap of the Higgs mode (c.f. [28, 29]).

## 4.3 Comparison with QMC

In this section, we compare the results of our QGW method with those from the full configuration interaction QMC calculations described in Sec. 3.2. Since the problem of a mobile impurity strongly interacting with a bosonic bath in a quantum critical regime is very challenging, such a comparison is highly useful. In particular, the QMC results can serve as a benchmark for the diagrammatic QGW method. As we shall see, our comparison also highlights differences between experiments realizing systems in the canonical and grand canonical ensembles. The study of finite systems is therefore important towards possible cross-benchmarking as discussed further in Sec. 5.

The QMC results presented in this section were obtained by optimizing a variational ansatz and using it for importance sampling in the simulation; see App B for details. The mean of the shift $S$ was used as the energy estimator. This was repeated for various lattice sizes ($M = 5 \times 5, 6 \times 6, \ldots, 10 \times 10$) and extrapolated to an infinite lattice size, as detailed in App. B.3. The QGW results are produced on a square lattice of size $M = 10 \times 10$ as in the previous sections, and the formalism works in the grand canonical ensemble in such a way that the density far away from the impurity is fixed to unity. In order to compare the results of the semi-analytical QGW method and the fully numerical QMC method, the hopping parameter employed in the latter has been rescaled by a factor $t_c/t_{\text{QMC}} = 0.7179$ to align the $O(2)$ critical points of the two methods [63]. The difference in critical points between the two methods reflects the mean-field nature of Gutzwiller ground state calculation. However, the QGW description of the quantum fluctuations is expected to be accurate as evidenced by the quantitative agreement found with compared to QMC in Ref. [28].

Before comparing the QGW and QMC results, we need to analyze a fundamental difference between the two approaches. The QMC simulations keep the number of bosons fixed and therefore work in the canonical ensemble. We find that for sufficiently attractive impurity-boson interactions $U_{\text{IB}}/U \lesssim -1$, the QMC calculations show that the impurity forms a bound state dimer with one boson in agreement with the QGW results discussed in Sec. 4.1.1. When there is one boson per site in a finite lattice, such a bound state, however, takes the system away from unit filling since one lattice site now contains the impurity and two bosons, see Fig. 3, making the effective filling fraction of the rest of the lattice $1 - 1/M$. This prevents the bosons from ever entering the MI phase, and instead the system is taken to the SF phase

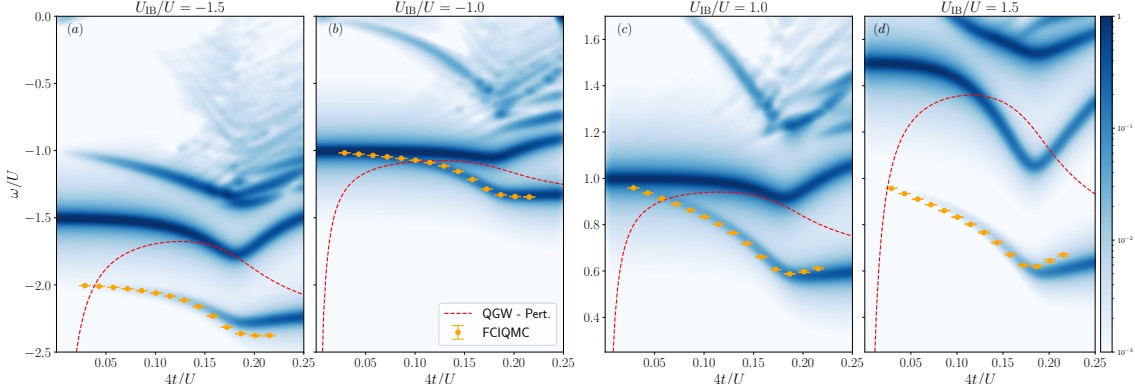

Figure 7: Impurity spectral function $A(\mathbf{k} = 0, \omega)$ from the field theoretical QGW method compared with the polaron energy from the QMC method (yellow points). The QGW results correspond to the filling $\langle \hat{n} \rangle = 0.99$ in panels $(a), (b)$, and $\langle \hat{n} \rangle = 1.01$ in panels $(c), (d)$, while the QMC results are extrapolated from the results for $M = (5 \times 5, 6 \times 6, \ldots, 10 \times 10)$ bosons on a square lattice of $M = N^2$ sites. The dashed red lines are second-order perturbation theory for the polaron energy from the QGW method [27], which unphysically diverges for $t/U \to 0$. This divergence is remedied by the ladder resummation. The QMC hopping parameter has been rescaled by a factor $t_c/t_{\mathrm{QMC}} = 0.7179$ to align the $O(2)$ critical points of the two methods.

just below the $\langle \hat{n} \rangle = 1$ Mott lobe for $t/U \to 0$ with an extremely small superfluid fraction. Likewise, QMC calculations show that for a strong and repulsive impurity-boson interaction $U_{\mathrm{IB}}/U \gtrsim 1$, the impurity pushes the bosons away from their lattice position, forming a bound state with a hole, again confirming the QGW results. This also takes the system away from unit filling to an effective filling fraction $1 + 1/M$ so that it stays just above the $\langle \hat{n} \rangle = 1$ Mott lobe. We have numerically verified that the QMC simulations do not enter the MI phase for $|U_{\mathrm{IB}}/U| \gtrsim 1$ by computing the charge gap across various values of $t$ and $U_{\mathrm{IB}}$ as explained further in App. B.4.

The QGW method, on the other hand, operates in the grand canonical ensemble where the bosonic bath maintains a constant density at a distance from the impurity, effectively treating the bosons as a particle reservoir. It should be noted however that the QGW method does not allow the impurity to change the density of the bath, and it interacts with a fixed bath. This means that the bath remains at unit filling even in the regime of strong impurity-boson interactions where bound states are formed, as we explicitly saw in Sec. 4.1.1. This fundamental difference between the two methods makes it necessary to adjust the filling of the QGW calculation to compare against results of the QMC method for strong interaction $|U_{\mathrm{IB}}/U| \gtrsim 1$.

First, Fig. 6(b) and (c) compare the results of the QGW and QMC calculations across the phase transition for weak interaction $|U_{\mathrm{IB}}/U| = \pm 0.5$. In this case, no adjustments to the QGW filling are necessary as no bound states are formed. We see that there is excellent quantitative agreement between the two methods.

In Fig. 7(a)-(b), we compare the QMC and QGW results for stronger attractive impurity-boson interactions. The QGW calculations are performed for a fixed non-integer filling $\langle \hat{n} \rangle = 0.99$ in order to compare with the QMC results in the presence of bound impurity-boson dimers, as explained above. This corresponds to a line in Fig. 1 very close to the $\langle \hat{n} \rangle = 1$ line in the superfluid phase up to the $O(2)$ point, after which it is just below the lower boundary of the $\langle \hat{n} \rangle = 1$ Mott lobe. Figure 7(b) demonstrates a remarkable quantitative agreement between the QGW and QMC calculations for the polaron ground state energy for $U_{\mathrm{IB}}/U = -1$.

This includes a highly non-trivial smeared cusp feature close to the phase transition point, which appears due to the inherited Mottness of the bath with decreasing $t/U$ despite never entering the insulating phase. For even stronger attraction $U_{\mathrm{IB}}/U = -1.5$ in Fig. 7(a), there is still good agreement between the QGW and QMC results. This confirms the accuracy of our diagrammatic QGW framework, which is remarkable given the very complex nature of a strongly correlated many-body system close to a quantum phase transition and given that the QGW is built upon a mean-field ansatz, which is unable to quantitatively predict the critical interaction strength for the MI to SF quantum phase transition.[5]

For completeness, in Fig. 7(a)-(b) we report the results from second-order perturbation theory, which completely fails to describe the polaron energy for strong interactions; see also Sec. 4.2. In particular, it predicts a diverging energy in the limit $t/U \to 0$ for non-integer filling. Here, the compressibility of the superfluid diverges [27], and the perturbative analysis predicts a divergence in the energy that comes from a macroscopic dressing cloud around the impurity. This prediction is unphysical since a non-zero $U$ still penalises a macroscopic dressing of the impurity. Our diagrammatic ladder resummation, on the other hand, provides a comprehensive description of this regime with a finite value for the ground state energy in excellent quantitative agreement with QMC. The ladder resummation therefore resolves the unphysical energy divergence predicted by perturbation theory, although the quasiparticle residue still vanishes as $t/U \to 0$. Instead for discussion of the orthogonality catastrophe in the context of the conventional Bose polaron problem, we direct the interested reader to Refs. [64, 65].

Figure 7(c)-(d) shows the same analysis performed for strong repulsive impurity-boson interactions.

The QGW calculations are performed using a filling fraction $\langle n \rangle = 1.01$ to account for the formation of bound impurity-hole states taking a finite system out of the MI phase, as explained above. Again, we observe excellent agreement between the QGW and QMC results, confirming the remarkable accuracy of our diagrammatic resummation, whereas perturbation theory completely fails to describe the system in this strongly correlated regime. For completeness, we have also compared the QGW and QMC in the superfluid regime at unit filling as a function of the interaction strength, and we find again an excellent quantitative agreement; see App. A.7 and Fig. 13.

Finally, we comment briefly on the extreme limit $U_{\mathrm{IB}}/U \to -\infty$. Here, we expect the impurity to form a cluster state with a macroscopic number of bosons in its dressing cloud. Such an $N$-body bound state cannot be described within our ladder approximation, which only includes two-body impurity-boson correlations, necessitating the inclusion of more diagrams or using variational wave functions as done for the conventional Bose polaron [9]. For large repulsive interactions $U_{\mathrm{IB}}/U \to \infty$, on the other hand, the impurity can still only push one boson away at unit filling, as shown in Fig. 3, and therefore we expect our ladder approximation to be reliable even in this extreme limit.

The fate of differences between the canonical and grand canonical ensembles in the thermodynamic limit is an interesting question. It should be noted that these effects have been explored in other contexts that show that impurities can significantly alter the bath [66].

# 5 Discussion and outlook

In this work, we explored a mobile impurity immersed in a strongly correlated lattice Bose gas in the vicinity of a $O(2)$ quantum phase transition between a MI and a SF phase at integer filling. Based on a QGW description of the bosonic bath, we developed a powerful field

---

[5]A similar feature has been reported for short range correlations when compared to QMC results in [28].

theoretical framework describing the impurity scattering with the fundamental excitations of the Bose gas. By resumming a selected class of generalized ladder diagrams, we showed how the interplay between strong boson-impurity interactions, quantum criticality, and the evolution from gapped particle-hole to Higgs and gapless Goldstone modes of the bosons gives rise to very rich and non-trivial physics with several polaron branches. Our semi-analytical field theory was furthermore shown to compare very well with quantum Monte-Carlo calculations, which is remarkable for such a strongly correlated many-body system. This demonstrates the utility of our field-theoretical framework and opens up several new research directions.

Our work highlights how polarons immersed in strongly correlated baths can exhibit much richer physics compared to more conventional polarons in weakly or non-interacting Bose and Fermi gases. It also illustrates how polarons can be used to probe non-trivial quantum many-body systems in the spirit of quantum sensing, as analyzed previously for example for geometric and topological properties of the environment [20–23,67]. The present results show that polarons can explore quantum criticality including the precise point of the phase transition. This motivates further investigations into how coherent superpositions of internal states of the impurity can be used to enhance the sensitivity of the impurity probe while minimizing the back-action on the environment [68–71].

The predictions of this work should be accessible in cold-atom experiments using optical lattices where the BH model and, in particular, the MI-SF transition have already been realized [72]. Radio-frequency pulses have been used in continuum atomic gases to measure the spectral function of polarons [9,73], and quantum gas microscopy in optical lattices can furthermore provide complementary information regarding the spatial properties of polarons [74,75]. This raises interesting questions concerning the wave function of the polaron and spatial correlations with the surrounding bosons in the quantum critical regime, which are left as the subject of future study. Polarons have also been observed in new 2D transition metal dichalchogenide semiconductors [76,77], which may open up ways to observe the predicted results in a solid-state setting.

Our theoretical framework can also be generalized to explore the properties of the polaron at non-zero temperature and momentum. Other interesting questions include the interaction between polarons mediated by the elementary excitations of the bosons in the quantum critical regime, which may support bound states (bi-polarons), as has been predicted for polarons in weakly interacting BECs [42,78,79]. Understanding this is crucial for developing a consistent quasiparticle description of a non-zero concentration of impurities in the BH model. Another interesting problem concerns a possible phase separation that takes place at the borders of the MI lobe for strong interactions [80].

Our results moreover reinforce the notion that an impurity, particularly in small systems, can significantly alter the bath's properties even taking it out of a given quantum phase. This raises questions concerning how to experimentally observe the sharp features predicted at unit filling in this paper. One could imagine using a setup where a harmonic trap creates a "wedding cake" structure of rings with different densities [81]. In such a setup, regions with a non-integer filling could act as particle reservoirs for the region with integer filling where the impurities are located, effectively realizing an impurity experiment with constant chemical potential. Taken together, these extensions show the potential for exploring a wide area of territory for new physics as well as cross-benchmarking theory and experiment in the field of quantum simulation [82].

## Acknowledgments

We thank Aleksi Julku, Arturo Camacho-Guardian, Nathan Goldman, Ivan Amelio, Chiara Menotti, Fabio Caleffi, and Chris Bradly for fruitful discussions.

**Funding information**    This work has been supported by the Danish National Research Foundation through the Center of Excellence "CCQ" (Grant agreement No. DNRF156), the Independent Research Fund Denmark- Natural Sciences via Grant No. DFF -8021-00233B. This research was supported in part by the National Science Foundation under Grant No. NSF PHY-1748958. This project has received financial support from Provincia Autonoma di Trento and the Italian MIUR through the PRIN2017 project CEnTraL (Protocol No. 20172H2SC4). The work was supported by the Marsden Fund of New Zealand (Contract No. MAU 2007), from government funding managed by the Royal Society of New Zealand Te Apārangi. We acknowledge support by the New Zealand eScience Infrastructure (NeSI) high-performance computing facilities in the form of a merit project allocation. This research used resources of the Oak Ridge Leadership Computing Facility at the Oak Ridge National Laboratory, which is supported by the Office of Science of the U.S. Department of Energy under Contract No. DE-AC05-00OR22725.

## A    Further details on QGW method

### A.1    Background

In this section, we provide further details on the QGW method relevant to this work (c.f. Refs. [28,28–30]). The QGW method is based on the canonical quantization of the Lagrangian

$$
\begin{aligned}
\mathfrak{L}[c, c^*] &= \left\langle \Psi_G \middle| i\hbar\, \partial_t - \hat{H}_B \middle| \Psi_G \right\rangle \\
&= \frac{i\hbar}{2} \sum_{\mathbf{r},n} [c_n^*(\mathbf{r})\dot{c}_n(\mathbf{r}) - \text{c.c.}] + J \sum_{\langle \mathbf{r},\mathbf{s}\rangle}[\psi^*(\mathbf{r})\psi(\mathbf{s}) + \text{c.c.}] - \sum_{\mathbf{r},n} H_n |c_n(\mathbf{r})|^2 ,
\end{aligned}
\tag{A.1}
$$

where $\hat{H}_B$ is the bath Hamiltonian in Eq.(1), and $H_n = U n(n-1)/2 - \mu n$. The Lagrangian is a functional of the complex amplitudes $c_n(\mathbf{r})$ of the Gutzwiller ansatz

$$
|\Psi_G\rangle = \bigotimes_{\mathbf{r}} \sum_n c_n(\mathbf{r})|n, \mathbf{r}\rangle .
\tag{A.2}
$$

The quantization promotes these amplitudes to operators that obey equal-time canonical commutation relations

$$
\left[\hat{c}_n(\mathbf{r}), \hat{c}_m^\dagger(\mathbf{s})\right] = \delta_{\mathbf{r},\mathbf{s}}\, \delta_{n,m} .
\tag{A.3}
$$

In analogy with the Bogoliubov approximation for a dilute Bose-Einstein condensate, these operators are expanded around the ground state values $c_n^0$ (see Sec. 3.1) as $\hat{c}_n(\mathbf{r}) = \hat{A}(\mathbf{r})c_n^0 + \delta\hat{c}_n(\mathbf{r})$ with local normalization operator $\hat{A}(\mathbf{r})$ and fluctuations $\langle\delta\hat{c}_n(\mathbf{r})\rangle = 0$, which can be expanded in momentum space as

$$
\delta\hat{c}_n(\mathbf{r}) \equiv M^{-1/2}\sum_{\mathbf{k}} e^{i\mathbf{k}\cdot\mathbf{r}}\,\delta\hat{C}_n(\mathbf{k}).
\tag{A.4}
$$

Retaining only up to quadratic terms in the fluctuations, one finds

$$
\langle\Psi_G|\hat{H}_B|\Psi_G\rangle \approx E_0 + \frac{1}{2}\sum_{\mathbf{k}}\left[\delta\hat{\underline{C}}^\dagger(\mathbf{k}, -\delta\hat{\underline{C}}(-\mathbf{k})\right]\hat{\mathcal{L}}_{\mathbf{k}}\begin{bmatrix} \delta\hat{\underline{C}}(\mathbf{k}) \\ \delta\hat{\underline{C}}(-\mathbf{k}) \end{bmatrix},
\tag{A.5}
$$

where where $E_0$ is the mean-field Gutzwiller ground-state energy, the vector $\delta \underline{\hat{C}}$ contains the components $\delta \hat{C}_n$, and $\hat{\mathcal{L}}_{\mathbf{k}}$ is a pseudo- Hermitian matrix [28]. Eq. (A.5) is diagonalized by a suitably chosen Bogoliubov rotation

$$\delta \hat{C}_n(\mathbf{k}) = \sum_\lambda u_{\lambda,\mathbf{k},n}\, \hat{b}_{\lambda,\mathbf{k}} + \sum_\lambda v^*_{\lambda,-\mathbf{k},n}\, \hat{b}^\dagger_{\lambda,-\mathbf{k}}, \tag{A.6}$$

recasts the quadratic form of the Hamiltonian into diagonal form

$$\hat{H}_B \approx \sum_\lambda \sum_{\mathbf{k}} \omega_{\lambda,\mathbf{k}}\, \hat{b}^\dagger_{\lambda,\mathbf{k}}\, \hat{b}_{\lambda,\mathbf{k}}, \tag{A.7}$$

where each $\hat{b}_{\lambda,\mathbf{k}}$ corresponds to a different many-body excitation mode with frequency $\omega_{\lambda,\mathbf{k}}$, labeled by its momentum $\mathbf{k}$ and index $\lambda$.

The accuracy of the QGW method can be estimated by calculating the magnitude of the quantum fluctuations about the Gutzwiller mean-field, quantified by the control function $F = 1 - \langle \hat{A} \rangle$. The control function $F$ displays a cusp at the $O(2)$ phase transition, as found in Refs. [28, 29]. However, in two dimensions, $F$ grows away from the transition in the limit $t/U$ due to the incorrect description of the condensate order parameter by the Gutzwiller mean-field ansatz as discussed in Ref. [29].

Contributions of quantum fluctuations of the bath beyond the mean field can be systematically included, as described in Ref. [28]. This procedure yields the expansion in Eq. (4). Here, the bath density operator has been expanded as $\hat{n}_{\mathbf{r}} \approx n_0 + \delta_1 \hat{n}(\mathbf{r}) + \delta_2 \hat{n}(\mathbf{r})$ where

$$\delta_1 \hat{n}(\mathbf{r}) = \sum_n n\, c_n^0 \left[ \delta \hat{c}_n(\mathbf{r}) + \delta \hat{c}_n^\dagger(\mathbf{r}) \right], \tag{A.8}$$

gives the Fröhlich-type coupling and

$$\delta_2 \hat{n}(\mathbf{r}) = \sum_n n\, \delta \hat{c}_n^\dagger(\mathbf{r})\, \delta \hat{c}_n(\mathbf{r}) - n_0\, \hat{\mathbb{1}} - \hat{A}^2(\mathbf{r}), \tag{A.9}$$

involves contributions from quantum fluctuations as well as their feedback onto the Gutzwiller mean-field ground state via the operator $\hat{A}(\mathbf{r})$. Application of the Bogoliubov rotation (Eq. (A.6)) to Eqs. (A.8) and (A.9) yields the vertex contributions present in Eq. (4) (see also Ref. [27]).

The QGW expansion of $\hat{H}_{\mathrm{IB}}$ (Eq. (4)) connects with the Bogoliubov approximation of $\hat{H}_{\mathrm{IB}}$ [42] in the limit $U/t \to 0$ where $|\psi_0|^2/n_0 \approx 1$ as discussed in Refs. [27, 28, 59]. Here, the canonical quantization procedure is applied instead to the condensate order parameter

$$\hat{\psi}(\mathbf{r}) = \sum_n \sqrt{n}\, \hat{c}_{n-1}^\dagger(\mathbf{r})\, \hat{c}_n(\mathbf{r}) \approx \psi_0 + \delta_1 \hat{\psi}(\mathbf{r}) + \delta_2 \hat{\psi}(\mathbf{r}) \ldots, \tag{A.10}$$

where $\psi_0$ corresponds to the mean-field condensate density and

$$\delta_1 \hat{\psi}(\mathbf{r}) = \frac{1}{\sqrt{I}} \sum_\lambda \sum_{\mathbf{k}} \left[ U_{\lambda,\mathbf{k}}\, e^{i\mathbf{k}\cdot\mathbf{r}}\, \hat{b}_{\lambda,\mathbf{k}} + V_{\lambda,\mathbf{k}}\, e^{-i\mathbf{k}\cdot\mathbf{r}}\, \hat{b}^\dagger_{\lambda,\mathbf{k}} \right], \tag{A.11}$$

while $\delta_2 \hat{\psi}(\mathbf{r})$ can be calculated by a straightforward extension. The quantities $|U_{\lambda,\mathbf{k}}|^2$ and $|V_{\lambda,\mathbf{k}}|^2$ are the quasi-particle and quasi-hole excitation strengths, respectively, and identify the character of elementary excitations [31]. In Ref. [59], $\hat{H}_B$ was expanded to order $\delta_1 \hat{\psi}(\mathbf{r})$ and the lowest collective mode was shown to become identical to the Bogoliubov dispersion in the $U/t \to 0$ limit. We note that in this limit, all other modes in the QGW method become strongly gapped, justifying the single-mode description.

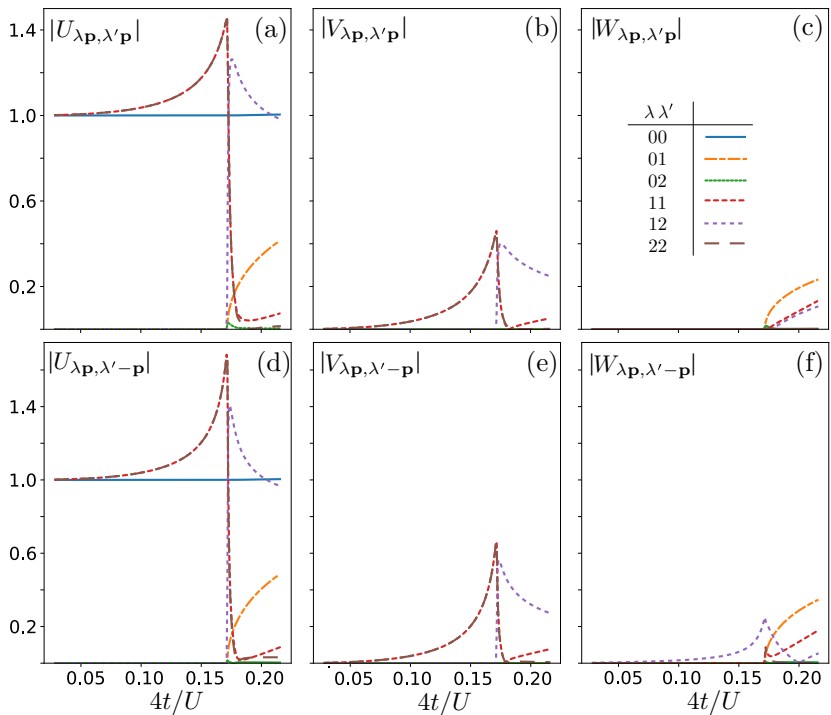

Figure 8: Magnitude of the interaction vertices across the $O(2)$ phase transition at unity filly. (a-c) interactions between equal momentum excitations. (d-f) interactions between opposite momentum excitations. Explicitly, $\lambda = 0$ refers to the Gutzwiller mean-field ground state, while $\lambda = 1, 2$ refers to particle and hole processes in the MI or Goldstone and Higgs processes in the superfluid phase, respectively. $\lambda\lambda' = nm$ refers then to the corresponding two-excitation process. Here we have evaluated the vertices at a fixed lattice momentum $|\mathbf{p}| = 0.39$ for clarity.

## A.2   Vertices across the $O(2)$ transition

In this section, we provide additional information about the two-particle vertices given in Eq. (5), which set the bare interaction strengths for the processes shown in Fig. 2(a). In general, these vertices have non-trivial momentum and mode dependence, particularly across the $O(2)$ transition where many processes determine the behavior of the lattice polarons.

In Fig. 8, we show results for the magnitudes of the vertices as a function of the hopping strength across the transition at unit filling. Here, we consider pair processes that involve elementary excitations of equal (a)-(c) as well as opposite (d)-(f) momenta. We remark that $\lambda = 0$ or $\lambda' = 0$ describes isotropic single-excitation processes involving scattering off the Gutzwiller mean-field.

We first discuss the MI regime. In (a), (b), and (d), (e), we see that the $U$ and $V$ vertices for particle-particle and hole-hole processes are equal (and opposite). We note from the form of the Bethe-Salpeter equation that only the $\Gamma_{22}^{\lambda\lambda}$ in-medium scattering matrix element contributes in this region, as shown in Fig. 2(c). There, the $V$ vertex contributes only to quantum corrections of the filling due to zero-point quantum fluctuations. The $W$ vertex instead describes the process exciting particle-hole pairs, which is followed by the ladder of $\hat{U}_3\hat{U}_3\ldots$ re-scatterings capped by $\hat{W}_3$, returning the pairs to the bath. We note that in the perturbative calculation at $\mathcal{O}(U_{\mathrm{IB}}^2)$, the effect of the $U$ vertex in processes is absent beyond the mean-field level. Rather, we see that this vertex plays a crucial role in determining the polarons via particle-particle or hole-hole processes depending on the sign of $U_{\mathrm{IB}}$ as discussed in Sec. 4.1.1. Here, we see that these processes have roughly equal (and opposite) strengths, which is a general charac-

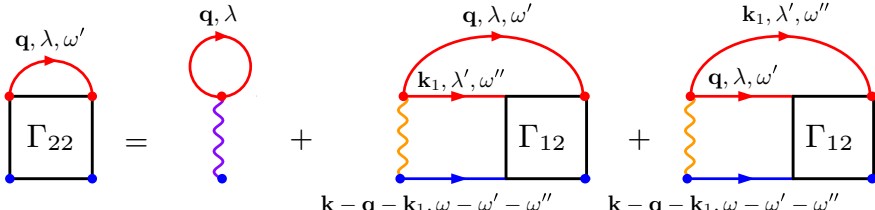

Figure 9: Loop diagram contribution to the impurity self-energy. Momentum and energy conservation is enforced, and the $\Gamma_{12}$ in-medium scattering matrix element is properly symmetrized according to Eq. (14). This results in two diagrams which give equal contributions to the self energy expression in Eq. (A.14), shown here for completeness.

teristic of the QGW calculation in the vicinity of the $O(2)$ point. In Figs. 8(c) and (f), we see instead that the $W$ vertex *only* contributes in the MI for particle hole pairs of opposite rather than equal momentum. This reflects the constraint of particle conservation on a single site in the MI in the generation of a local particle-hole pair, which recalls the physics producing the particle-hole dressing cloud of the conventional Fermi polaron (c.f. [7]).

On the superfluid side of the $O(2)$ point, we see instead that $W$ vertices for equal momentum become non-zero, while the $U$ and $V$ vertices inheriting particle-particle and hole-hole processes from the MI sharply drop off, with visible tradeoff to particle-hole (Higgs-Goldsone) pair processes on the superfluid side of the transition. Notably, a cusp is formed as a result of this tradeoff, which is reflected as a non-analyticity in the polaron energies across the transition, as noted in Sec. 4.2. Physically, this tradeoff reflects an abrupt change in the processes which produce the dressing cloud of the polarons. Namely, the cloud is dominated by the virtual excitation of Higgs and Goldstone modes, which are generated at all orientations of pair momentum. Notably, at zero temperature the $V$ vertices do not contribute beyond $\mathcal{O}(U_{\text{IB}})$ and are therefore absent from the ladder summations.

Single-particle processes, which vanished in the MI, now rapidly become non-zero across the phase transition. In particular, this consists of processes initiated by $\hat{W}_2$ or $\hat{U}_2$ followed by re-scatterings via $\hat{U}_3$ and capped by $\hat{W}_1$ or $\hat{U}_1$. Here, we see that the corresponding vertices are dominated by the Goldstone mode. We comment on the limit $U/t \to 0$ and the connection with the Bogoliubov approach, which has been discussed previously in Refs. [28,59]. In this limit, the Gutzwiller ansatz is well-described by the discrete Gross-Pitaevskii equation (DGPE), with the ground state well-approximated by a coherent state. Furthermore, the excitation spectrum is described by the Goldstone mode, which recovers the well-known Bogoliubov dispersion with good agreement for the sound velocity in the $t/U \to 0$ limit. Instead, the Higgs mode, which is not captured in the DGPE, is increasingly gapped with $t/U$.

### A.3 Explicit expressions for the self-energy

In this section, we provide explicit expressions for the zero-temperature self-energy diagrams shown in Fig. 2(c) and discussed in Sec. 3.1.3. From this figure, we find five distinct contributions to the self-energy

$$\Sigma(\mathbf{k},\omega) = \sum_{i,j=1}^{2} \Sigma_{ij}^{00}(\mathbf{k},\omega) + \sum_{\lambda>0} \Sigma_{22}^{\lambda\lambda}(\mathbf{k},\omega). \tag{A.12}$$

The first four contributions follow simply from the in-medium scattering matrix as $\Sigma_{ij}^{00}(\mathbf{k},\omega) = \Gamma_{ij}^{00}(\mathbf{k},\omega)$ and therefore can be obtained from Eq. (14). The fifth contribution

however requires the summation over the closed loop of the elementary excitation as

$$\Sigma_{22}^{\lambda\lambda}(\mathbf{k},\omega) = -\int \frac{d\omega'}{2\pi} \sum_q D_\lambda^{(0)}(\mathbf{k}-\mathbf{q},\omega') \Gamma_{22}^{\lambda\lambda}(P,\mathbf{q},\mathbf{q}), \tag{A.13}$$

where $P = (\mathbf{k}, \omega - \omega')$ and $D_\lambda^{(0)}$ is the Green's function for a bosonic bath excitation defined in Sec. 3.1.1. The Feynman diagram corresponding to Eq. (A.13) is shown in Fig. 9, where we sum over indistinguishable processes involving the $\Gamma_{12}$ in-medium scattering matrix element according to the action of the symmetrizer in Eq. (14) and the Feynman rule discussed in Sec. 3.1.2. This produces a trivial factor of 2.

Crucially, we note that the collisional energy of the $\Gamma_{12}$ in-medium scattering matrix element is just $\omega$, as the process described involves two incoming elementary excitations at energies $\omega'$ and $\omega''$ as well as the incoming impurity at energy $\omega - \omega' - \omega''$, which matches the total energy $\omega$ of the outgoing impurity line in the calculation of the impurity Green's function. This diagram gives then the following contribution to the self-energy

$$\sum_\lambda \Sigma_{22}^{\lambda\lambda}(\mathbf{k},\omega) = U_{IB} \sum_{\mathbf{q}} \sum_\lambda V_{\lambda q, \lambda q} + \frac{U_{IB}}{M} \sum_{q,\mathbf{k}_1} \sum_{\lambda,\lambda_1 > 0} \frac{W_{\lambda q, \lambda_1 k_1} \Gamma_{12}^{\lambda_1 \lambda}(K, \mathbf{k}-\mathbf{k}_1, \mathbf{k}+\mathbf{q})}{\omega - \omega_{q,\lambda} - \omega_{\mathbf{k}_1,\lambda_1} - \varepsilon_{P-q-k_1} + i0^+}, \tag{A.14}$$

where $K = (\mathbf{k}, \omega)$. We recognize the first contribution in Eq. (A.14) as $U_{IB}\langle \delta_2 \hat{n}\rangle$, the quantum correction to the filling due to zero-point quantum fluctuations of the bath. We note that iteration to quadratic order in the coupling produces the perturbative expression found in Ref. [27]

$$\begin{aligned}\Sigma(\mathbf{k},\omega) \approx{}& U_{IB}\langle\hat{n}\rangle + \frac{U_{IB}^2}{\sqrt{M}} \sum_{k,\lambda} \frac{|W_{k\lambda,0,0} + U_{k\lambda,0,0}|^2}{\omega - \omega_{k,\lambda} - \varepsilon_{q-k} + i0^+} \\ &+ \frac{U_{IB}^2}{2M} \sum_{\substack{k,p \\ \lambda,\lambda_1}} \frac{|W_{\lambda k, \lambda_1 p}|^2}{\omega - \omega_{k,\lambda} - \omega_{p,\lambda_1} - \varepsilon_{q-k-p} + i0^+},\end{aligned} \tag{A.15}$$

where we note differing conventions for the vertices used in that work. The dashed red lines in Figs. 4-7 correspond to the results for $\Sigma(\mathbf{0}, 0)$ at this level of approximation.

## A.4 Finite-size effects

In this section, we discuss finite-size effects in the QGW method calculation of the impurity spectral function. For simplicity, we analyze results within the non-self-consistent approximation (NSC), where the simple replacement discussed in Sec. 3.1.4 is not made. In this approximation, finite-size effects are more easily interpreted, and the limitations of the NSC approximation, alluded to in Sec. 3.1.4, become visually evident. In lattice systems, the grid spacing of the reciprocal lattice is determined by the size of the system with $k_{n_\lambda} = 2\pi n_\lambda/\sqrt{M}$, $n_\lambda = 0, 1, \ldots, \sqrt{M} - 1$. The closed-loop integrations in the Bethe-Salpeter equation (Eq. (14)) run over the grid of the reciprocal lattice, and therefore the impurity-bath scattering continua are only finitely resolved. This is illustrated in Fig. 10 for lattice sizes $M = 6^2$ and $10^2$, where the ($\mathbf{P} = 0$) single-excitation continua across the $O(2)$ transition for the lowest three collective modes are shown. Here, the lines correspond to fixed values of the crystal momentum, with the density of lines of a particular color determined by the corresponding local density of states. Furthermore, the width of the continua for each band broadens with increasing $t/U$ as anticipated from the scaling of the widths of $\varepsilon_\mathbf{k}$ and the Bogoliubov mode. We see then that finite-size effects become more pronounced in this limit as the bandwidth of the single-excitation continua increases while the momentum spacing in the first Brillouin zone

remains fixed. Therefore, it becomes numerically prohibitive to study lattice polarons in the deep superfluid regime within the QGW method, where instead the Bogoliubov theory becomes numerically advantageous [42]. This is further justified by the growth of the gaps for all but the lowest collective mode, which justifies the neglect of these modes in the deep superfluid regime. We note that these remarks apply also to two-excitation continua, which also enters the calculation of the self-energy through the final diagram of Fig. 2(c).

In Fig. 11, we show finite-size effects in the in-medium scattering matrix calculated within the NSC approximation for lattice sizes $M = 6^2$ and $10^2$. Here, we note two classes of features: (i) layered horizontal bands and (ii) curves attached to the boundaries of these banded regions asymptoting with $\pm U_{\mathrm{IB}}/U$. Features (i) are the single-excitation continua, filling in with increasing system size. This increased resolution also corresponds to a defining of the upper and lower boundaries of the continua; see Fig. 3. In particular, the lower boundary establishes the scattering threshold, while the upper boundary is set by the finite bandwidth of the particular mode. We note that for a calculation including $N_{\mathrm{mode}}$ modes, there are $N_{\mathrm{mode}}$ distinct single-excitation continua as well as $N_{\mathrm{mode}}(1 + N_{\mathrm{mode}})/2$ two-excitation continua. However, their "weight" in Fig. 11 over regions of $\omega/U$ is ultimately determined by the vertices which enter into the closed-loop integrations in the Bethe-Salpeter equation (Eq. (14)). Furthermore, both impurity-Goldstone and impurity-Higgs single-excitation continua are visible in Fig. 11, which can be confirmed by visual comparison with Fig. 10. We then understand the resolution of features of type (i) as the finite-size effect discussed in the previous paragraph. However, features of type (ii) show finite-size effects for $|U_{\mathrm{IB}}/U| \ll 1$ where they merge with the various scattering thresholds. These are the energies of the upper and lower impurity-bath bound states, which include impurity particle and impurity hole states that are expected to be weakly bound for $|U_{\mathrm{IB}}/U| \ll 1$, hence extended and sensitive to finite size effects. Away from the single-excitation continua however, these lines are robust to finite-size effects, consistent with a strongly bound, localized state. We comment that for attractive couplings, the linear scaling $\propto U_{\mathrm{IB}}/U$ is consistent with the asymptotic energy $U_{\mathrm{IB}} + 4t$ of the vacuum impurity-particle

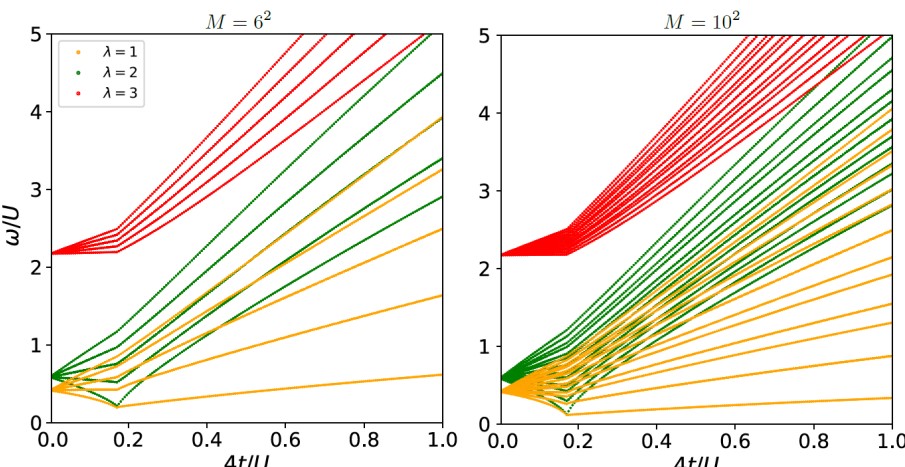

Figure 10: Single-excitation impurity scattering continua along the $\langle \hat{n} \rangle = 1$ line for system sizes (a) $M = 6^2$ and (b) $M = 10^2$. The continua associated with elementary excitations $\lambda = 1$ (orange), 2 (green), and 3 (red). In the superfluid regime, the $\lambda = 1$ mode corresponds to the Goldstone mode, while the $\lambda = 2$ modes corresponds to the Higgs mode with the latter displaying a visible non-analytic cusp in the opening and closing of the gap at the $O(2)$ transition. Continuous lines correspond to a fixed value of the lattice momentum as a function of $4t/U$ with line density reflecting the local density of states.

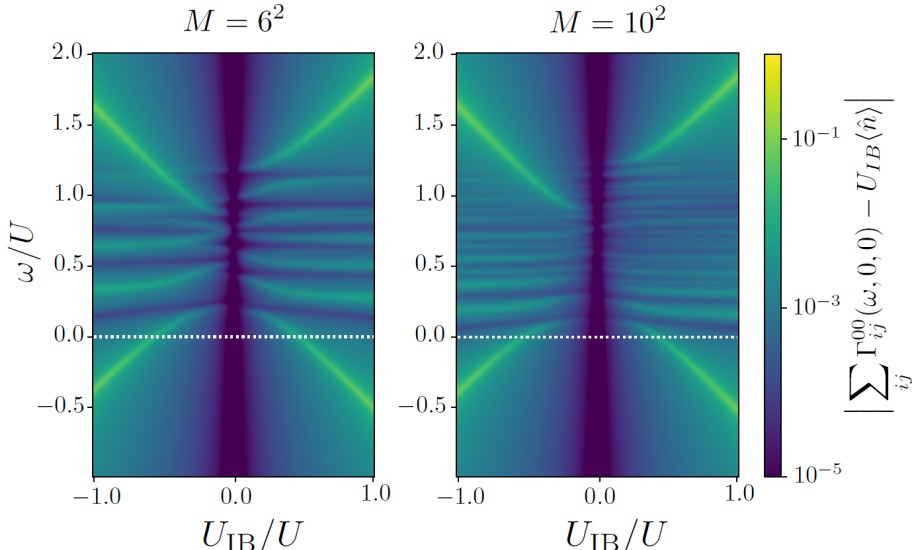

Figure 11: In-medium scattering matrix evaluated for $\mu/U = 0.4142$ and $4t/U = 0.1723$ for system sizes (a) $M = 6^2$ and (b) $M = 10^2$, corresponding to the superfluid edge of the $O(2)$ transition. The zero-momentum components of the in-medium scattering matrix are shown for simplicity, which describes scattering between the impurity and particles in the mean-field ground state. Additionally, we subtract the mean-field shift $U_{\text{IB}}\langle \hat{n} \rangle$, which is fixed and does not display finite-size effects.

bound state energy for $|U/U_{\text{IB}}| \rightarrow 0$ [42, 61, 83]. The impurity-hole bound state, including its upper branch, has no vacuum limit because it requires the concept of a hole. However, these states obey an analogous linear scaling law with the coupling to the impurity-particle bound states due to particle-hole symmetry.

## A.5 General Bethe-Salpeter equation

For completeness, we give the Bethe-Salpter equation at finite temperatures in Fig. 12. This equation can be written self consistency using a matrix notation:

$$\underline{\underline{\Gamma}}(P,p,p') = \underline{\underline{V}}(p,p') + \underline{\underline{V}}(p,p')\underline{\underline{\Pi}}(P,p_1)\underline{\underline{\Gamma}}(P,p,p_1), \tag{A.16}$$

where $\underline{\underline{\Gamma}}$ and $\underline{\underline{V}}$ are indicated explicitly in Fig. 12 while the pair propagator matrix is given by:

$$\underline{\underline{\Pi}} = \begin{pmatrix} \xrightarrow{\hspace{1cm}} & 0 \\[1mm] \xrightarrow{\hspace{1cm}} & \xleftarrow{\hspace{1cm}} \\[1mm] 0 & \xrightarrow{\hspace{1cm}} \end{pmatrix}$$

Figure 12: Diagrammatic representation of the general Bethe-Salpeter equation. In the zero temperature limit, the expression in Fig. 2b is recovered.

We note that $\underline{\underline{\mathbf{\Pi}}}_{22}$ vanishes in the zero temperature limit from which the expressions in Sec. 3.1.2 can be recovered. Furthermore, the presence of internal back-propagating excitation lines requires the equivalent representations of the $\hat{W}_1$ and $\hat{W}_2$ processes discussed in Sec. 3.1.1.

## A.6 Solving the Bethe-Salpeter equation

In this section, we detail the numerical method utilized to solve the Bethe-Salpeter equation (Eq. (14)) in the NSC approximation. As discussed in Sec. 3.1.2, $\Gamma_{11}$ and $\Gamma_{12}$ can be obtained by matrix inversion, with the remaining elements of the in-medium scattering matrix following then straightforwardly via matrix multiplication. Without loss of generality, we then detail how to obtain $\Gamma_{11}^{\lambda\lambda'}(P, \mathbf{p}, \mathbf{p}')$ in the case $\mathbf{P} = 0$ considered in this work. First, we work on a grid of momentum $k_{n_\lambda}$ discussed in Sec. A.4, writing the in-medium scattering matrix elements as six-dimensional arrays, which must be re-evaluated for each value of $z$. The Bethe-Salpeter equation for $\Gamma_{11}$ becomes

$$
\begin{aligned}
(\Gamma_{11}[z])_{p_{i_x}, p_{j_y}, p_{n_x}, p_{m_y}}^{\lambda\lambda'} &= U_{\mathrm{IB}} U_{p_{i_x}, p_{j_y}, p_{n_x}, p_{m_y}}^{\lambda\lambda'} \\
&+ \frac{U_{\mathrm{IB}}}{\sqrt{M}} \sum_{\lambda_1} \sum_{n', m'} \frac{U_{p_{i_x}, p_{j_y}, p_{n'_x}, p_{m'_y}}^{\lambda\lambda_1}}{z - \omega_{p_{n'_x}, p_{m'_y}, \lambda_1} - \varepsilon_{p_{n'_x}, p_{m'_y}}} (\Gamma_{11}[z])_{p_{n'_x}, p_{m'_y}, p_{n_x}, p_{m_y}}^{\lambda_1\lambda'}.
\end{aligned}
\tag{A.17}
$$

This can be written as an element of the matrix equation given by Eq. (A.16):

$$
\underline{\underline{\mathbf{\Gamma}}}_{11}[z] = \underline{\underline{\mathbf{V}}}_{11} + \underline{\underline{\mathbf{V}}}_{11} \underline{\underline{\mathbf{\Pi}}}_{11}[z] \underline{\underline{\mathbf{\Gamma}}}_{11}[z],
$$

where

$$
(\underline{\underline{\mathbf{V}}}_{11} \underline{\underline{\mathbf{\Pi}}}_{11}[z])_{p_{i_x}, p_{j_y}, p_{n_x}, p_{m_y}}^{\lambda\lambda'} = \frac{U_{\mathrm{IB}}}{\sqrt{M}} \frac{U_{p_{i_x}, p_{j_y}, p_{n'_x}, p_{m'_y}}^{\lambda\lambda_1}}{z - \omega_{p_{n'_x}, p_{m'_y}, \lambda_1} - \varepsilon_{p_{n'_x}, p_{m'_y}}}.
$$

This matrix equation can then be solved by inversion $\underline{\underline{\mathbf{\Gamma}}}_{11}[\omega + i\eta] = (\underline{\underline{\mathbf{1}}} - \underline{\underline{\mathbf{\Pi}}}_{11}[\omega + i\eta])^{-1} \underline{\underline{\mathbf{V}}}_{11}$ where $\eta$ is a positive infinitesimal number taken to be $\eta = 10^{-2}$ in this work. Furthermore, these are square matrices with $N_{\mathrm{band}}^2 M^4$ total elements. Here, $N_{\mathrm{band}}$ is the cutoff on the number of modes included in the calculation, which determines the summations of $\lambda$, i.e. $\lambda = 0, 1, \ldots, N_{\mathrm{band}} - 1$, where we recall that $\lambda = 0$ denotes the mean-field Gutzwiller ground state. In this work, we take $N_{\mathrm{band}} = 3$, which leads to converged results in all the regimes considered. Furthermore, the Fock space summations are cutoff at $N_{\mathrm{Fock}} = 7$, noting that this is sufficient for numerical convergence in the quantum critical regime but must eventually be increased in the limit $U/t \to 0$ where the ground state becomes a coherent state [59]. With these cutoffs and grids, the calculation of the lattice polaron self energy can be accomplished on the timescale of a few hours or less on a standard machine.

## A.7 Self-consistent approximations

In this section, we discuss the implementation of the self-consistent (SC) approximation within the QGW method discussed in Sec. 3.1.4. Because the vertices and multi-band excitation spectrum are calculated numerically in the QGW method, the implementation of the SC approximation has a more significant numerical cost than in the Bogoliubov method where the vertices and mode spectra are known analytically and the ground state is assumed coherent. However, this cost is associated with the strongly correlated nature of the BH bath in the quantum critical regime, as opposed to a weakly interacting BEC bath. Multiple iterations and successive improvements of the in-medium scattering matrix elements $\Gamma_{(n),ij}^{\lambda\lambda'}$ and self-energies $\Sigma_{(n)}$, then

quickly become numerically prohibitive, motivating the mean-field implementation taken in the main text.

In this section, we compare various implementations of the SC approximation as shown in Fig. 13. In Fig. 13(a), the NSC result is shown, where we see a polaron line following the mean-field energy at attractive couplings. At repulsive couplings, this line enters, and decays into, the impurity-bath continua and is replaced by an energy following the impurity-hole binding energy from Fig. 11. The merging of the polaron line with the continuum is analogous to what was found in the NSC treatment of Ref. [42].

In Fig. 13(b), we show results for a single iteration ($n = 1$) of the SC approximation. Here, we see that the NSC impurity-bath continua now depend on the impurity-bath interaction strength, and the line of highest spectral density matches the QMC result. However, we see that this is not the ground state as there are faint spectral lines for repulsive coupling at lower energy. In comparison to panel (a), we see that these are remnants of the NSC dimer lines, which is a clear artifact of the iterative procedure. Namely, the NSC result for the self-energy has poles at these locations, which consequently also appear as poles in subsequent iterations. Further iterations can reduce these effects; however, as a consequence, a continued fraction of poles is produced in the propagator, which requires many iterations to dampen.

In the present work, the numerical cost of iterating on the order of $10^1$ times to investigate the convergence of this procedure was found to be numerically prohibitive. Instead, we have implemented the back action in a minimal way by incorporating the mean-field energy in the bare impurity dispersion in the calculation of the Bethe-Salpeter equation as described in Sec. 3.1.4. This produces the result in Fig. 13(c), which is in good quantitative agreement with the QMC prediction and without remnants. We note that this simple approximation

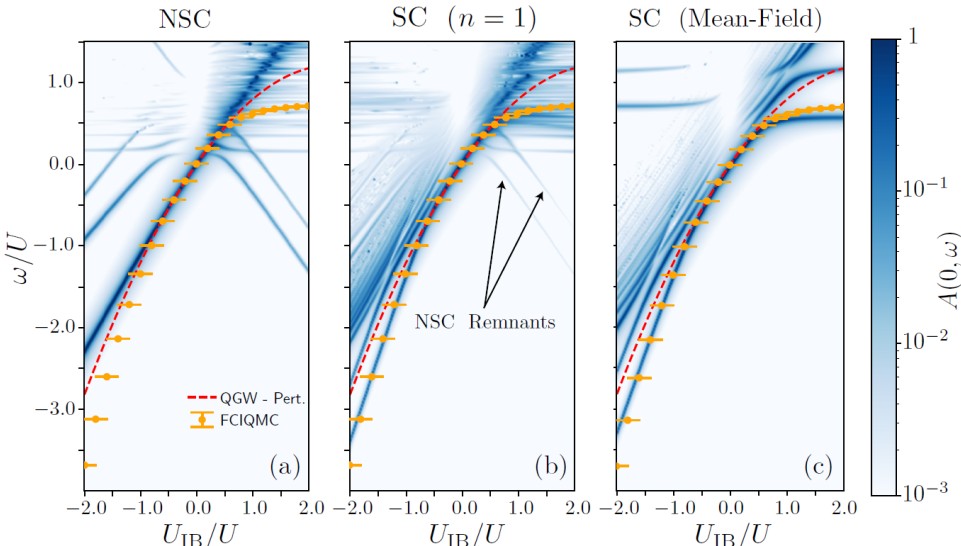

Figure 13: Impurity spectral function $A(\mathbf{k} = 0, \omega)$ from the QGW method within different levels of approximation for $\mu/U = 0.4014$, $4t/U = 0.2154$, and unit filling $\langle \hat{n} \rangle = 1$ for a square lattice with $M = 10^2$ sites. (a) Results of the NSC approximation with coupling independent impurity-bath continua thresholds. The region of largest spectral density follows (roughly) the perturbative QGW polaron energy indicated by the red dashed line. (b) Results of the ($n = 1$) SC approximation, where remnants of the NSC continua and bound state spectrum from (a) are indicated explicitly. (c) Results of the NSC approximation with mean-field shifted impurity dispersion. The QMC results are shown for comparison with hopping parameter rescaled by a factor $t_c/t_{\mathrm{QMC}} = 0.7179$ to align the $O(2)$ critical points of the two methods.

appears to provide good agreement with the QMC method in the coupling regimes considered in this work; however, its utility for larger coupling strengths where the back action effects are possibly more significant is questionable.

# B   Further details on the full configuration interaction QMC method

In this section, we discuss the details of the full configuration interaction QMC computations. We start by discussing the setup required to make the Hamiltonian of Eq. (1) compatible with full configuration interaction QMC. Next, we discuss the importance sampling scheme we used to reduce full configuration interaction QMC's inherent statistical bias. Finally, we discuss the extrapolation scheme we used to extrapolate our results to infinite lattice sizes.

## B.1   Matrix representation of the Hamiltonian

To be able to apply the full configuration interaction QMC algorithm, we start by representing the Hamiltonian of Eq. (1) as a matrix in the basis of Fock states (here for a single species of bosons)

$$|\mathbf{n}\rangle = |n_1, n_2, \dots, n_M\rangle = \prod_{i=1}^{M} \frac{1}{\sqrt{n_i!}} \left(\hat{a}_i^\dagger\right) |0\rangle, \tag{B.1}$$

where $n_i$ is the number of bosons occupying the lattice site $i$ and $M$ is the number of lattice sites (and for the purpose of this representation, all lattice sites of a given square lattice are labeled by a scalar index). Now, the Hamiltonian can be realized as a matrix $\mathbf{H}$ with the matrix elements

$$H_{\mathbf{n},\mathbf{m}} = \langle \mathbf{n}|\hat{H}|\mathbf{m}\rangle. \tag{B.2}$$

Similarly, a quantum state can be written as

$$|\psi\rangle = \sum_{\mathbf{n}} c_{\mathbf{n}} |\mathbf{n}\rangle, \tag{B.3}$$

where $c_{\mathbf{n}}$ are elements of the coefficient vector $\mathbf{c}$. When an impurity is present in addition to the many-particle boson bath, we use product states of a Fock basis for the bosons and for the impurity as a basis. Now, the matrix realization of the Hamiltonian can be used as shown in Eq. (16), allowing us to estimate the energy and properties of the ground state of $\hat{H}$.

Realizing the BH Hamiltonian in a Fock basis yields a matrix that blocks with respect to the number of particles. Starting a QMC computation with a vector $\mathbf{c}^{(0)}$ with a fixed particle number will only sample that block in $\mathbf{H}$. This makes the chemical potential $\mu$ a simple energy shift and allows us to set it to zero. Moreover, the BH Hamiltonian in the real-space Fock basis is stoquastic (i.e. all off-diagonal matrix elements are negative) and thus full configuration interaction QMC does not experience a sign problem. This allows us to treat relatively large systems without additional approximations. All QMC calculations in this work were performed with the `Rimu.jl` package [58].

## B.2   Importance sampling

To reduce the noise and population control bias in full configuration interaction QMC, we apply importance sampling [55–57]. Importance sampling is a similarity transform applied to the Hamiltonian $\mathbf{H}$

$$\tilde{\mathbf{H}} = \mathbf{\Phi}\mathbf{H}\mathbf{\Phi}^{-1}, \tag{B.4}$$

where $\boldsymbol{\Phi}$ is the diagonal matrix with elements taken from a guiding vector $\boldsymbol{\phi}$

$$\Phi_{\mathbf{n},\mathbf{n}} = \phi_{\mathbf{n}}. \tag{B.5}$$

While transforming $\mathbf{H}$ this way does not change its spectrum, it transforms the eigenvectors. In particular, if $\mathbf{v}$ is an eigenvector of $\mathbf{H}$, the corresponding eigenvector of $\tilde{\mathbf{H}}$ is $\boldsymbol{\Phi}\mathbf{v}$. If $\boldsymbol{\phi}$ is close to the ground state eigenvector of $\mathbf{H}$, the modified eigenvector becomes more compact, which has the beneficial effect of reducing noise in the algorithm, which in turn reduces population control bias and makes computations more efficient [54,57]. However, it should be noted that $\boldsymbol{\Phi}$ does not need to be a particularly good approximation of the ground state to benefit the computation. As such, it is common to use a simple ansatz to use as the guiding vector [57].

In this work, we use

$$\phi_{\mathbf{n}}(p) = \left( \prod_{i=1}^{M} \frac{1}{n_i!} \right)^p, \tag{B.6}$$

for the elements of the guiding vector, where $n_i$ is the occupation number of site $i$ in the Fock basis state $|\mathbf{n}\rangle$ and $p$ is a parameter that can be optimized. This is the exact ground-state eigenvector of the one-component non-interacting BH model when $p = 1/2$, and it equally describes the Mott insulating ground state obtained for $t = 0$ when $p \to \infty$. We have found that by varying $p$, it can give reasonable estimates of the ground state even when the strength of the interaction is high. We have also observed that optimizing this ansatz on a smaller system and using the same value of $p$ for a larger one still significantly improves the performance of QMC. In the case of the larger systems, we were unable to even finish computations without importance sampling, as they would require more memory than what was available.

## B.3 Computation parameters and extrapolation

We compute the energies presented in this paper using the following procedure. First, we optimize the importance sampling parameter $p$ on a $3 \times 3$ lattice by minimizing the variational energy of the guiding vector. We can do this since, benefiting from importance sampling, the ansatz does not need to be optimized perfectly and an approximate setting is good enough [57]. Then, we sample an initial vector $\mathbf{c}^{(0)}$ from the optimized ansatz using a kinetic Monte Carlo procedure [84]. This is done to reduce QMC equilibration times. Finally, we run QMC for 100 000 steps, where we discard the first 25 000 and use the rest to compute a sample mean of the shift $S^{(n)}$, which gives an estimate of the ground state energy of the system. The standard error of the energy is estimated from the variance of the time series after removing

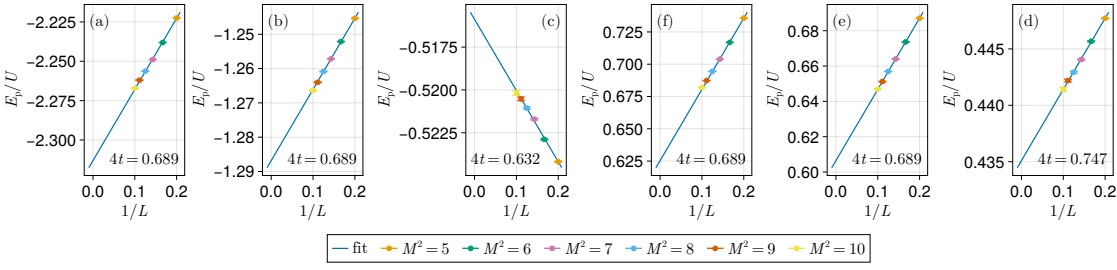

Figure 14: Examples of the application of the extrapolation procedure. The solid line is the fitted curve (Eq. (B.8)) and the dots represent QMC results for various lattice sizes with error bars. The panes (a-f) show fits for $U_{\mathrm{IB}}/U = -1.5, -1, -0.5, 0.5, 1, 1.5$, respectively.

correlations by a blocking analysis [85] with automated hypothesis testing [86]. The population control bias is estimated using the methods of Ref. [54] and found to be smaller than the stochastic standard error. All QMC results in this work are presented with error bars, which, however, are typically smaller than the marker size in the plots.

To compute the polaron energy $E_p$, we run two separate computations, one with the interacting impurity, which gives us the ground state energy $E^{(U_{IB} \neq 0)}$, and another with a non-interacting impurity (that is, setting $U_{IB} = 0$ in Eq. (1)), giving the ground state energy $E^{(U_{IB}=0)}$. Then, we calculate the polaron energy for a given lattice with $M$ sites as

$$E_p^{(M)} = E^{(U_{IB} \neq 0)} - E^{(U_{IB}=0)}. \tag{B.7}$$

For each pair of parameters $(U_{IB}, t)$, we compute the finite system polaron energy $E_p^{(M)}$ on grids of $M = 5^2, 6^2, \ldots 10^2$ sites. The result is then extrapolated to infinite system size by fitting $E_p$ and $\lambda$ to the relation

$$E_p^{(M)} = E_p + \lambda \frac{1}{\sqrt{M}}, \tag{B.8}$$

where $M$ is the number lattice points. The extrapolated energy $E_p$ is reported as the QMC result in the main part of the paper. Some examples of the extrapolation procedure are shown in Fig. 14.

## B.4 The charge gap

To verify that a strongly interacting impurity disturbs the bath and prevents it from entering the MI phase, we compute the charge gap for different values of the impurity strengths $U_{IB}$ as a function of the hopping strength $t$. The charge gap is defined as

$$\Delta E_c = \frac{1}{2} \left( E_{N=M+1} - E_{N=M-1} - 2E_{N=M} \right), \tag{B.9}$$

where $E_N$ is the energy of the ground state with $M$ lattice sites and $N$ bosons in the bath.

In the Bose-Hubbard model, the charge gap is an order parameter for the MI-SF phase transition. In an infinite system, it has a value of zero in the SF phase and non-zero in the MI phase [87]. We compute the charge gap for lattices of sizes $M = 4^2, 6^2, 8^2$ and $10^2$. The data is presented in Fig. 15. As seen in the figure, the computed charge gap values at strong impurity-boson interaction ($|U_{IB}/U| \geq 1$) are small and decrease with system size, which is consistent with them being zero for an infinite system. At zero hopping, $\Delta E_c$ can be computed analytically and is zero if and only if $|U_{IB}/U| \geq 1$, regardless of the size of the system.

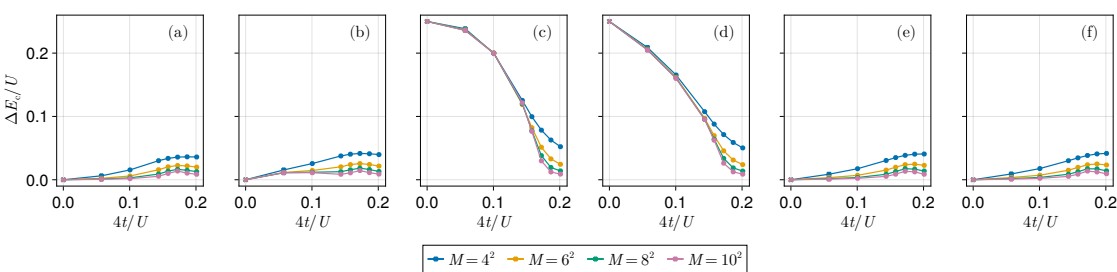

Figure 15: Disappearance of the charge gap for strong impurity-boson interaction. The plot panes (a-f) show the values of the charge gap $\Delta E_c$ for $U_{IB}/U = -1.5, -1, -0.5, 0.5, 1, 1.5$, respectively, at various lattice sizes $M$. The error bars on the points are smaller than the symbol size and the lines are guides to the eye. The values shown at $4t/U = 0$ are computed analytically.

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
