# Peer review of "Lattice Bose polarons at strong coupling and quantum criticality"

_SciPost Physics, doi:SciPost Phys. 19, 002 (2025)_

## Round 1 · Referee Report · Pietro Massignan (Referee 1) · 2025-4-20

Report

The manuscript by R. Alhyder et al. discusses the properties of mobile impurities interacting strongly with a bosonic lattice gas, which is tuned across the SF-MI quantum phase transition.

The paper is very well-written, the diagrammatic and QMC calculations are challenging but explained in detail, the graphs are illustrative, and the results appear correct. Furthermore, the considered set-up may be engineered in currently available experiments. For all these reasons, I can certainly recommend the publication of this manuscript in Scipost Physics.

Requested changes

Here below I list some remarks and comments, that the authors could implement in a revised version of the manuscript:

1 - the acronym QGW for "Quantum Gutzwiller" is somehow strange, given that "Gutzwiller" is a single word; i.e., what is the letter "W" standing for?

2 - in the abstract, and also on line 617: "resuming" ---> "resumming"

3 - the word "Mottness" may be familiar to some specialists, but is certainly not of general use. As such it sounds weird, specially when used in the Abstract. I suggest to rephrase that sentence, removing this word, or at least explaining it better.

4 - in Fig.1, it should be clarified that the two lowest panels compose Fig.1b. Else, call 1b the left panel, and 1c the right one.

5 - regarding Fig.1b(left), it is unclear to me why the lowest branch is called "particle", and the upper one "hole". How do the two excitations differ?

6 - in Eq.(1), I believe that a "minus" sign is missing in front of the first $\sum$ sign (i.e., it should be $-t$, rather than $t$)

7 - on line 102 there is a wrong link: Fig. 3.1.1(a) ---> Fig. 1(a)

8 - in Sec. 3.1 (or in Sec. 4 at the latest), I would specify somewhere that calculations are performed with $\lambda = 0,1,2$ at most. As of now, this is only stated in one of the Appendices.

9 - lines 142-144: could the authors explain more in detail with $\mu'(t)<0$ means "hole SF", while $\mu'(t)>0$ means "particle SF"?

10 - line 145: interfering ---> interacting

11 - I noticed that more than once Figures appeared much before the page where they were referenced, making it hard for the reader to follow the narration (in such a long and complex paper). Figures should appear on the page where they are referred to, or after that.

12 - $M$ is undefined in Eq.(14). Only much later the reader learns that $M$ is the total number of sites (i.e., the system volume)

13 - on line 264: "considering a superfluid fraction" ---> "considering a condensate fraction"

14 - in Fig. 4, the red line plays a key role, but its explicit formula is hidden inside Ref.[27]. I suggest to quote the explicit formula giving the red line (2nd order perturbation result) in this paper as well.

15 - line 377: "is a finite-size" ---> "are a finite-size"

16 - line 432: "superfluid" ---> "condensate"

17 - The relation between Figs. 6 and 7, and a good part of discussion in Sec. 4.3 were unclear to me. I understand that QMC works in the canonical (at fixed number of particle), while QGW is in the grand-canonical, this is clear. But why Fig.6a is so different from 7a, when the filling has changed by just 1%? (for example, why do the two red lines differ so much?) Moreover, it is difficult to compare the various panels in figures 6 and 7. Maybe the two figures may be joined in a single one, containing a grid of 4*2 panels? (if so, then the same values of U_{IB}/U may be used in panels b and c of Figs. 6 and 7)

18 - regarding the orthogonality catastrophe (OC): in the caption of Fig.7 it is said that the ladder resummation "remedies" the OC, and similarly on line 589 the word "resolves" is used. Notice however that in Ref. [64] a very different message is passed: in the context of usual Bose polarons, the authors of [64] showed that the OC must be present when the Bose bath becomes ideal (i.e., non interacting), and that a ladder treatment is clearly not able to recover it (while a GPE approach can). Notice furthermore that QMC calculations are normally performed with ~100 particles, and become inaccurate exactly in the OC limit, where the size of the dressing cloud explodes. For a discussion of finite-size effects in QMC, see for example the discussion around the vertical arrows in Fig. 4 of [N. Yegovtsev et al., Phys. Rev. A 110, 023310 (2024)]

19 - lines 609-610: shouldn't all differences between canonical and grand-canonical disappear in the thermodynamic limit?

20 - line 627: "as have been analyzed" ---> "as analyzed"

21 - some references are duplicated: for example [7] and [83], or [20] and [66]

22 - check the spelling of the initials of the second author of Ref. [63]

23 - check Ref. [60]: "and O. U."?

24 - Eq. (A.1): clarify how is $H_B$ related to $H$ given in Eq.(1). Similarly, it is unclear how Eq. (2) relates to Eq. (1)

25 - caption of Fig.8: "evaluate a" ---> "evaluated the vertices at a"

Recommendation

Publish (easily meets expectations and criteria for this Journal; among top 50%)

---

## Round 1 · Referee Report · Anonymous (Referee 2) · 2025-4-22

Report

In the present manuscript, the authors investigate the problem of an impurity particle immersed in an interacting Bose gas in a square lattice.

To describe the Bose medium, they use the quantum Gutzwiller method previously introduced by some of the authors to investigate the Bose-Hubbard model. Considering short range interactions between the impurity and the bath particles, the authors derive the relevant vertices in terms of the Bogoliubov operators describing the collective excitations of the medium obtained using the above mentioned method.

Then, they introduce a diagrammatic theory for the polaron self energy where the impurity-medium excitation scattering is treated in a ladder approximation, which goes beyond the recent perturbative theory introduced by some of the authors in Ref. [27].
In particular, the ladder diagrams capture the presence of bound states between the impurity and a medium particle.

Focusing on the case where the medium state is in the vicinity of the O(2) critical point between Mott-insulator and superfluid phase, the authors use their theory to calculate and analyze the impurity spectral function across the transition.
In addition, they present comparisons between the results of their diagrammatic approach and the ground state obtained from Monte Carlo simulations, and find a remarkable agreement.

I found the paper clearly written and the results rather well explained.
The results presented are original and seem correct. In my opinion, the present manuscript may be suitable publication once the authors have considered my remarks below.

Requested changes

Please find below my comments:

  • It is maybe a naive question, but could the authors comment on the reason they use of a finite size system for the calculation based on 'quantum Gutzwiller' method? Is it not possible to consider the homogeneous and infinite case (which, I believe has been used to obtain Figure 1)?

  • The definition of the vertices in Eq. 5) involve sums over n. I presume that these sums are truncated at some n_max, could the authors comment on how they choose the value of such n_max?

  • Eqs. 7) and 10) contain the same products of operators and are represented by the same diagrams in Fig. 2a). Similarly Eqs. 6) and 11) are made of identical products of operators, but the corresponding diagrams in Figure 2a) differ, is this normal or is there a typo somewhere?

  • In Section 4.3, the authors compare the results of their Gutzwiller diagrammatic approach with Monte Carlo calculations. In particular, the authors explain that in order to compare the regime of strong impurity-bath interactions, they need to adjust the density to a non-integer filling density (and hence outside the Mott phase) for the Gutzwiller method. I wonder if a comparison could be done in the Mott phase if the MC simulations were performed with (M+1) or (M-1) bosons instead of M bosons?

  • The authors briefly mention that the Gutzwiller method is inaccurate for the determination of the O2 critical point. Could the authors elaborate on the limitations of the method and in particular on how this could affect the polaron properties?

  • In the appendix A2, the authors refer to particle-particle and hole-hole processes that are equal and opposite in the Mott-phase. Do these correspond to the case with $\lambda\lambda'=11 $ and $ 22$? Since the figure 8 shows many lines, it may be beneficial for the reader to be explicit.

  • I find equation A12) a bit confusing. I could not find the definition of $D^{(0)}$ nearby, and perhaps the sum over $k$ should instead be over $\bf q$?

  • It seems that a factor $U_{IB}$ may be missing in the second term in Eq. A13)?

  • It seems that the footnote p.8 and the sentence line 174-175 repeat themselves.

  • line 490 typo: 'has been have been'

Recommendation

Publish (meets expectations and criteria for this Journal)

---

## Round 2 · Referee Report · Pietro Massignan (Referee 1) · 2025-5-29

Report

The Authors have responded appropriately to all my remarks, and in my opinion also to the ones raised by the other Referee, and they have suitably modified the manuscript. For this reason, I certainly recommend publication of this revised version in SciPost Physics.

Recommendation

Publish (easily meets expectations and criteria for this Journal; among top 50%)

---

## Round 2 · Referee Report · Anonymous (Referee 2) · 2025-6-5

Report

I believe that the authors have addressed most of my comments appropriately.

In my opinion, their updated manuscript deserves publication in SciPost Physics.

Minor comment:
I realize that my previous remark regarding the diagrams shown in Fig.2a and Eqs. (6-13) was probably not very clear.
I understand that the underlying processes for the U and W terms in Eq. 5 differ.
My remark was that I was wondering about the reason why the \lambda lines in the diagrams for U1 and W2 go in opposite directions while the ones in the U2 and W1 diagrams go in the same direction?
I could not understand this difference by simply looking at Eqs. (6-13).

Recommendation

Publish (easily meets expectations and criteria for this Journal; among top 50%)

---

## Round 2 · Author Response

Dear Editor,

We are very grateful for the careful reading, insightful comments, and positive feedback of the Referees. Their comments have been very valuable for improving the manuscript and provided opportunity to clarify passages which previously assumed specialized or specific knowledge on the QGW method. We hereby resubmit our manuscript along with a detailed response to both referees and a manuscript in which the changes were indicated in blue color text. We hope that referee 2 finds their remarks satisfactorily addressed such that they find it suitable to recommend for publication in SciPost Physics.

Sincerely yours,

The Authors

---

## Round 2 · List of Changes

\noindent Note: All the changes to the manuscript differing from the previous version appear in {\color{blue}blue}.

%%%%%%%%%%%%%%%%%%%%%%%%%%%%%%%%%%%%%%%%%%%%%

Letter to Referee 1

We thank the referee for their careful reading of our manuscript, their many insightful questions, and their positive feedback. In particular we are grateful for their questions regarding comparisons between QMC and QGW, which have lead to revisions highlighting and calrifying the subtleties of comparing the two methods. We hope that with our responses below and the corresponding revisions and clarifications added to the manuscript, the referee finds their remarks sufficiently well addressed to recommend for publication in SciPost Physics.

Kind regards,
The Authors

\subsection*{Detailed response to Referee 1}

\referee{Comment 1: It is maybe a naive question, but could the authors comment on the reason they use of a finite size system for the calculation based on 'quantum Gutzwiller' method? Is it not possible to consider the homogeneous and infinite case (which, I believe has been used to obtain Figure 1)?}

\rep{In general, and especially near the MI/SF phase transition, the ground state, excitations, and vertices in the QGW method must be calculated numerically. Lacking analytic expressions, unlike in the Bogoliubov treatment, the infinite case can only be numerically approximated by considering finite lattices. For calculations of the phase diagram and low-lying excitations, lattices with more than 100 sites are numerically realistic. Once the two-excitation vertices are required, which are essentially 6-dimensional arrays\footnote{Take for example $U_{\lambda{\bf k},\lambda'{\bf k}'}$ which has four momentum and two mode indices for a square lattice.}, the computational cost increases considerably. This is why a $10\times10$ lattice was used for the QGW calculations in the present work with finite-size effects discussed in App.~A.4.}

\sep

\referee{Comment 2: The definition of the vertices in Eq. 5) involve sums over $n$. I presume that these sums are truncated at some $n_{max}$, could the authors comment on how they choose the value of such $n_{max}$?}

\rep{Indeed, the summations are truncated at an $n_{max}$ suitably chosen to ensure the calculation is well converged. In practice, the value of $n_{max}$ within and in the vicinity of the unity filling MI can be kept small as the ground state is composed of only a few Fock states, while it must be made larger as the hopping is increased and the ground state becomes a coherent state. In practice, we have used $n_{max}=7$ and verified convergence for the points on the phase diagram considered in our work. We have added a footnote in the first paragraph of Sec.~V (also to address Comment 8 of referee 2) where the parameters of the QGW numerics are given.}

\sep

\referee{Comment 3: Eqs. 7) and 10) contain the same products of operators and are represented by the same diagrams in Fig. 2a). Similarly Eqs. 6) and 11) are made of identical products of operators, but the corresponding diagrams in Figure 2a) differ, is this normal or is there a typo somewhere?}

\rep{We thank the referee for pointing out a convention that we have chosen, which may indeed be somewhat confusing. As indicated in the paragraph beginning at line 174, we have chosen not to combine these processes even though they contain identical products of operators. This was done for two reasons. The first reason is for practicality: This choice allows a {\it single} Bethe-Salpeter matrix equation to be written as in Eq.~(14) and Fig. 2b. Indeed, Eq.~(14) is valid for all values of $\lambda$, $\lambda'$ (including the Gutzwiller mean-field ground state for one or both indices), which was a nontrivial task. The second reason is that they have different interaction strengths which can be seen by inspecting Eq.~(5). The utility of keeping these vertices disctinct can be seen from Fig. 8, where we see the different scalings of the interaction strengths with $4t/U$.}
%and (not shown) different threshold laws with the momentum ${\bf p}$.}

\sep

\referee{Comment 4: In Section 4.3, the authors compare the results of their Gutzwiller diagrammatic approach with Monte Carlo calculations. In particular, the authors explain that in order to compare the regime of strong impurity-bath interactions, they need to adjust the density to a non-integer filling density (and hence outside the Mott phase) for the Gutzwiller method.
I wonder if a comparison could be done in the Mott phase if the MC simulations were performed with $(M+1)$ or $(M-1)$ bosons instead of $M$ bosons? }

\rep{The referee raises an interesting idea, which is related to a subtle difference between QGW and MC. Let us for concreteness focus on strong repulsive interactions $U_{IB}>U$ and unit filling $\langle\hat n\rangle=1$, i.e.\ one impurity and $M$ bosons. Here, the polaron energy should go to $U$ for $t/U\to 0$ corresponding to the impurity having pushed a boson away from its site so that a site with two bosons is created. This is essentially what both the QGW and MC give.
Had we instead performed MC calculations with $M-1$ bosons, the energy would go to zero for $t/U\to 0$ corresponding to one particle per site throughout the lattice: $M-1$ sites with a boson each and one site with the impurity. We have in response to the referee performed numerical calculations, which indeed confirm this.
So the QGW and MC calcuations would not agree using this procedure since the limiting states are qualitatively different, and a comparison would not make sense.

What is measured of course depends on the precise experimental conditions, which is why we have made an effort to carefully explain the conditions of our results. Since this is a recurrent point raised by the referees, and to avoid confusion, we added one sentence in Sec. 4.3 to state that the problem is solved in the grand canonical to compute the bath properties, but the impurity interacts and modifies the bath locally without changing its global properties.
}

\sep

\referee{Comment 5: The authors briefly mention that the Gutzwiller method is inaccurate for the determination of the O2 critical point. Could the authors elaborate on the limitations of the method and in particular on how this could affect the polaron properties?}

\rep{Indeed, the different critical points between QMC and Gutzwiller is a reflection of the mean-field nature of the ground-state calculation in the latter approach. The QGW description of quantum fluctuations however is accurate as evidenced by quantitative agreement between QGW and QMC for polaron properties in the present work, and for correlation functions of the bath (in Ref.~[28]) once the critical point is rescaled. Therefore, we do not expect the critical point mismatch to limit the QGW description of the polaron w.r.t. QMC. We note that the Gutzwiller prediction of the critical point is expected to be brought into agreement with QMC following an iterative calculation where quantum fluctuations are accounted for self-consistently in the calculation of the ground state. We added additional further elaboration on this point to the second paragraph of Sec.~4.3.}
\sep

\referee{Comment 6: In the appendix A2, the authors refer to particle-particle and hole-hole processes that are equal and opposite in the Mott-phase. Do these correspond to the case with $\lambda\lambda'=11$ and $22$? Since the figure 8 shows many lines, it may be beneficial for the reader to be explicit.}

\rep{The referee is correct. We have added additional text to the caption of Fig.~8 to guide the reader.}

\sep

\referee{Comment 7: I find equation A12) a bit confusing. I could not find the definition of $D_{(0)}$ nearby, and perhaps the sum over $k$ should instead be over $q$? }

\rep{We thank the referee for catching this. The internal summation is indeed over $q$, while $D^{(0)}_\lambda$ is the Green's function for the bosonic excitation $\lambda$ of the bath.}

\sep

\referee{Comment 8: It seems that a factor $U_{IB}$ may be missing in the second term in Eq. A13)?}

\rep{We thank the referee for the very careful reading of the manuscript. This mistake is now
corrected}

\sep

\referee{Comment 9: It seems that the footnote p.8 and the sentence line 174-175 repeat themselves.}

\rep{We have removed the footnote and added text to lines 174-175.}

\sep

\referee{Comment 10: Line 490 typo: 'has been have been'}

\rep{Corrected.}

\sep

%%%%%%%%%%%%%%%%%%%%%%%%%%%%%%%%%%%%%%%%%%%%%%%%%%%%%%%%%%%%

Letter to Referee 2

We thank the referee for their many insightful questions, very careful reading and highlighting of typos, and praise and high recommendation of the work. In particular, we are grateful to the referee for highlighting unclear passages, where we have taken their feedback into account very seriously in our revisions. We address the requested changes of the referee below and hope that they find their remarks well addressed.

Kind regards,
The Authors

\subsection*{Detailed response to Referee 2}

\referee{Comment 1: The acronym QGW for "Quantum Gutzwiller" is somehow strange, given that "Gutzwiller" is a single word; i.e., what is the letter "W" standing for?}

\rep{We appreciate this point made by the referee, but the Gutzwiller trial wave function can be found referred to in the literature for the Hubbard model with the `GW' acronym (see for instance Phys. Rev. Lett. 86, 2605 (2001)). When the QGW model was introduced in Ref.~[28], this convention was adopted and a `Q' was added.}

\sep

\referee{Comment 2: In the abstract, and also on line 617: "resuming" ---> "resumming"}

\rep{We thank the referee for spotting this mistake, which is now corrected}

\sep

\referee{Comment 3: The word "Mottness" may be familiar to some specialists, but is certainly not of general use. As such it sounds weird, specially when used in the Abstract. I suggest to rephrase that sentence, removing this word, or at least explaining it better.}

\rep{We use ``Mottness'' simply to refer to the influence of Mott physics on the bosonic bath. Following the referee's recommendation, we have rephrased this sentence in the abstract to be (hopefully) less technical.}

\sep

\referee{Comment 4: In Fig.1, it should be clarified that the two lowest panels compose Fig.1b. Else, call 1b the left panel, and 1c the right one.}

\rep{We have now pulled the figure labels out of the plots, and updated the caption to indicate that (b) refers to both the plots in the bottom row.}

\sep

\referee{Comment 5: Regarding Fig.1b(left), it is unclear to me why the lowest branch is called "particle", and the upper one "hole". How do the two excitations differ?}

\rep{The ``hole" or ``particle" character of a mode refers to the relative weights of the quasi-particle $(U_{\lambda,{\bf k}})$ and quasi-hole $(V_{\lambda,{\bf k}})$ amplitudes in the expansion of the order parameter fluctuations in terms of the elementary excitations as shown in Eq.\ (A11).
In the SF phase, both amplitudes can be non-zero whereas
one of these amplitudes is non-zero and the other zero in the Mott phase corresponding to a pure particle or hole mode.
Analytic expressions for the particle and hole branches can be found derived in Ref. [58] where this identification is discussed further (also Ref. [34]). We have added additional text surrounding Eq. (A11) highlighting its significance in the interpretation of the character of various branches.}

\sep

\referee{Comment 6: In Eq.\ (1), I believe that a "minus" sign is missing in front of the first $\Sigma$ sign (i.e.\ it should be $-t$, rather than $t$)}%

\rep{Corrected}

\sep

\referee{Comment 7: On line 102 there is a wrong link: Fig. 3.1.1(a) ---> Fig. 1(a)}

\rep{Fixed}

\sep

\referee{Comment 8: In Sec. 3.1 (or in Sec. 4 at the latest), I would specify somewhere that calculations are performed with $\lambda=0,1,2$ at most. As of now, this is only stated in one of the Appendices.}

\rep{We have in response to the referee added a footnote to the first paragraph of Sec.\ 4 clarifying the mode and occupation number cutoffs used in the calculation. We chose not to add these details to Sec. 3 as the formalism presented in that section is completely general. It is important to clarify that calculations were performed with $\lambda = 0,1,2$ not because including $\lambda =3$ is prohibitive but rather we find our results are unchanged by the inclusion of $\lambda =3$, indicating convergence with respect to the number of modes. In practice, the modes for $\lambda>2$ are increasingly gapped and do not contribute to the results. Additionally, we have checked for convergence with respect to occupation numbers, which has informed the corresponding cutoff. We do not show these results in our work, however such checks are standard (and assumed) in the literature.}

\sep

\referee{Comment 9: Lines 142-144: could the authors explain more in detail with $\mu'(t)<0$ means "hole SF", while $\mu'(t)>0$ means "particle SF"?}

\rep{
In solid state physics one distinguishes between electron or hole conductance depending on the band filling. The chemical potential decreases (increases) by increasing the band width ($t$) for a particle (hole) state. As an example, one can just consider a 1D band $\mu=-4t \cos(k_F)$. The same occurs for a superfluid close to the Mott lobe although the effective Mott band filling is not the same as for free fermions (hole superfluidity requires generally $n>0.5$). Such an analogy and the name ``hole superfluidity'' has been pointed out already in the seminal paper Fisher et. al., Phys. Rev. B 40, 546 (1989), and thoroughly discussed in the context of solitons and vortices (see the review [31]) and elementary excitations [34]. Although this topic is not the main focus of our paper, we have added additional explanation and citations to the relevant paragraph in Sec.~3.1.}

\sep

\referee{Comment 10: Line 145: interfering ---> interacting}

\rep{Corrected}

\sep

\referee{Comment 11: I noticed that more than once Figures appeared much before the page where they were referenced, making it hard for the reader to follow the narration (in such a long and complex paper). Figures should appear on the page where they are referred to, or after that.}

\rep{We thank the referee for pointing out this formatting issue. We will work with the Scipost Physics team to make sure that this is improved in the published version. Currently, the figures are placed in the appropriate sections in our teX file.}

\sep

\referee{Comment 12: $M$ is undefined in Eq.(14). Only much later the reader learns that $M$ is the total number of sites (i.e., the system volume)}

\rep{We agree with the referee and have added the definition of $M$ to the first paragraph of Sec.~2.}

\sep

\referee{Comment 13: On line 264: "considering a superfluid fraction" ---> "considering a condensate fraction"}

\rep{Corrected.}

\sep

\referee{Comment 14: In Fig. 4, the red line plays a key role, but its explicit formula is hidden inside Ref.[27]. I suggest to quote the explicit formula giving the red line (2nd order perturbation result) in this paper as well.}

\rep{We thank the referee for pointing out this unintended omission. We have added the explicit formula to Appendix~A.3.}

\sep

\referee{Comment 15: Line 377: "is a finite-size" ---> "are a finite-size"}

\rep{Corrected}

\sep

\referee{Comment 16: Line 432: "superfluid" ---> "condensate"}

\rep{Corrected}

\sep

\referee{Comment 17: The relation between Figs. 6 and 7, and a good part of discussion in Sec. 4.3 were unclear to me. I understand that QMC works in the canonical (at fixed number of particle), while QGW is in the grand-canonical, this is clear. But why Fig.6a is so different from 7a, when the filling has changed by just $1\%$? (for example, why do the two red lines differ so much?)
Moreover, it is difficult to compare the various panels in figures 6 and 7.
Maybe the two figures may be joined in a single one, containing a grid of 4*2 panels? (if so, then the same values of $U_{IB}/U$ may be used in panels b and c of Figs. 6 and 7)}

\rep{We thank the referee for this insightful question.
Figure 6 is for a unit filling fraction $\langle\hat n\rangle=1$ so that the reservoir undergoes a MI-SF quantum phase transition at a critical value of $t/U$, whereas Fig.\ 7 is for $\langle\hat n\rangle=0.99$ so that the reservoir is always SF, staying just below the $\langle\hat n\rangle=1$ Mott lobe for $t/U\rightarrow 0$. This is the basic reason for the differences between the two figures. In particular, the red lines differ substantially because an unphysical divergence in the energy incorrectly predicted by perturbation theory for $t/U\to0$ for non-integer filling. For integer filling $\langle\hat n\rangle=1$ in Fig.\ 6, we moreover only show QMC results for
small $U_\text{IB}/U$ because the impurity binds/repels one of the bosons for larger $|U_\text{IB}|/U$ making the effective filling fraction of the surrounding bath non-integer.
Numerically, we see this by a vanishing charge gap as discussed in App. B.2 and Fig. 15. When this happens, we cannot compare the QMC calculations with the QGW results, since the latter fix the density of the reservoir to $\langle\hat n\rangle=1$.
Finally, we have opted not to plot six values of $U_{IB}/U$ in Fig. 6 and Fig. 7 in order to not overload the reader but instead provide the most relevant results. Additionally, the choice was made to focus on parameter regimes where QMC and QGW could be compared.
We have now emphasized this more in the text.

To make this clearer we have changed the Fig. 7 caption and improved the discussion in Sec.\ 4.3.
}

\sep

\referee{Comment 18: Regarding the orthogonality catastrophe (OC): in the caption of Fig.7 it is said that the ladder resummation "remedies" the OC, and similarly on line 589 the word "resolves" is used. Notice however that in Ref. [64] a very different message is passed: in the context of usual Bose polarons, the authors of [64] showed that the OC must be present when the Bose bath becomes ideal (i.e., non interacting), and that a ladder treatment is clearly not able to recover it (while a GPE approach can).
Notice furthermore that QMC calculations are normally performed with ~100 particles, and become inaccurate exactly in the OC limit, where the size of the dressing cloud explodes. For a discussion of finite-size effects in QMC, see for example the discussion around the vertical arrows in Fig. 4 of [N. Yegovtsev et al., Phys. Rev. A 110, 023310 (2024)]}

\rep{
We thank the referee for raising this subtle point. There is in fact no contradiction between the results of the present paper and those in Ref.~[64].
In Ref.\ [64], the Bose bath is made ideal by letting the interaction constant $g \to 0$. The phonon energy then vanishes and the condensate can reorganise over large distances involving a macroscopic number of bosons once the impurity is introduced in the system.
This gives rise to the bosonic orthogonality catastrophe (OC). A simple ladder approximation cannot capture this physics since it only allows the dressing of the impurity by maximally one boson as discussed in Ref.\ [64].

In our work by contrast, the on-site repulsion $U$ between the bosons is kept non-zero while the hopping $t \to 0$. Here, the compressibility diverges because the single-particle bandwidth vanishes so that any change of the chemical potential in the system introduces a macroscopic number of particles, not because interactions vanish. A non-zero $U$ still penalises extended density rearrangements, so the impurity dresses itself with only a finite number of bosons even when $t/U\to 0$. The energy divergence that appears in second order perturbation theory is therefore a truncation artifact and not physical. Summing the ladder diagrams self-consistently re-introduces the scale $U$ to infinite order and yields a finite ground-state energy that matches our QMC data all the way to $t/U\to0$. In this sense, it “remedies” the energy divergence in this lattice limit.

To make this clearer we have changed the Fig. 7 caption and improved the discussion in Sec.\ 4.3, adding also the suggested reference.

}

\sep

\referee{Comment 19: Lines 609-610: shouldn't all differences between canonical and grand-canonical disappear in the thermodynamic limit?}

\rep{
The referee raise a subtle and interesting question. While the canonical and grand canonical ensembles agree regarding average values in the thermodynamic limit, they do not predict the same fluctuations.
In particular, the number of particles is fixed in the canonical ensemble whereas it can vary in the grand canonical ensemble. Since the impurity can push an extra mobile boson/hole out to the surrounding lattice, this can take the system out of the MI as explained in our response above to
comment 17, although the conductivity goes to zero in the thermodynamic limit.

}
\sep

\referee{Comment 20: Line 627: "as have been analyzed" ---> "as analyzed"}

\rep{Corrected}

\sep

\referee{Comment 21: Some references are duplicated: for example [7] and [83], or [20] and [66]}

\rep{Corrected.}

\sep

\referee{Comment 22: Check the spelling of the initials of the second author of Ref. [63]}

\rep{Fixed.}

\sep

\referee{Comment 23: Check Ref. [60]: "and O. U."?}

\rep{Fixed.}

\sep

\referee{Comment 24: Eq. (A.1): clarify how is $H_B$ related to $H$ given in Eq.(1). Similarly, it is unclear how Eq. (2) relates to Eq. (1)}

\rep{The passage of the bath Hamiltonian from Eq. (1) to (2) is achieved via an expansion around the Gutzwiller ground state and truncation to quadratic order in fluctuations followed by a Bogoliubov rotation as detailed in App.~A.1. This procedure is thoroughly detailed in previous works [PhysRevResearch.2.033276], and we have added citations to these works surrounding Eq. (2) to direct the interested reader. Furthermore, we have added a few of the intermediate steps to App. A.1.}

\sep

\referee{Comment 25: Caption of Fig.8: "evaluate a" ---> "evaluated the vertices at a"}

\rep{Corrected}

\sep

---

## Round 3 · Author Response

We thank the referee 1 for this clarifying comment. As to the particular orientations of $\hat{W}_1$ and $\hat{W}_2$, we note that they can be represented in two (equivalent) ways, due to the symmetry of $W$-vertex with dummy label exchange. I.e. either with the appropriate excitation line on the left or the right hand side of the vertex depending on the context. Furthermore, we note that there are two contributions to $\hat{W}_1$ and $\hat{W}_2$, which can be seen by by taking $\lambda=0$ or $\lambda'=0$ of the first and second $W$-term in Eq.~(4), yielding the factor 2 in Eqs.~(10) and (11).

In practice, however we agree with the referee that it is more intuitive to show the diagrams in a way that can be employed for the zero-temperature ladder summations used throughout the text. Therefore, we have changed the labeling of the $W$-processes such that $\hat{W}_1$ matches the orientation of $\hat{U}_1$ and likewise for $\hat{W}_2$ and $\hat{U}_2$. This can be seen in the updated Fig. 2a. and Eqs. (10) and (11) and is reflected throughout the text. Furthermore, we have added additional text on lines 126, 174, 181, 441, and 770. In particular, the addition on line 126 clarifies that the coefficients $v_{\lambda, {\bf k}, n}$ and $u_{\lambda, {\bf k}, n}$ can be chosen to be real, which makes evident the properties of the vertices mentioned above. We hope that with this more intuitive diagrammatic representation, it is clarified for the reader how the processes can be linked to construct diagrams describing different zero-temperature observables.

For completeness, we note that in App.~A.5, Fig.~12 the alternative (equivalent) diagrammatic representations of the $\hat{W}_1$ and $\hat{W}_2$ processes are needed to evaluate the general Bethe-Salpeter equation due to the possibility of back-propagating bath excitation lines. We have added additional text in line 855 bringing this to the reader's attention.

---

## Round 3 · List of Changes

Summary of changes Here we detail changes to the manuscript that were made in addition to those discussed already above. 1. Line 738: The word `isotropic' was inserted to describe processes where the momentum orientation is unimportant. 2. Figure 2: The vertex $W_1$ was changed to show the variation which makes it more consistent with the overall formalism. 2. Figure 8: The caption was updated to clarify that the magnitude of the lattice momentum was held fixed in all subplots.

---

## Editorial Decision

published